# Contemporary Surgical Management of Colorectal Liver Metastases

**DOI:** 10.3390/cancers16050941

**Published:** 2024-02-26

**Authors:** Pratik Chandra, Greg D. Sacks

**Affiliations:** 1Department of Surgery, NYU Grossman School of Medicine, New York, NY 10016, USA; pratik.chandra@nyulangone.org; 2VA New York Harbor Healthcare System, New York, NY 10010, USA

**Keywords:** colorectal liver metastasis, treatment sequencing, portal vein embolization, resection margin, hepatic arterial infusion pump, two-stage hepatectomy, associating liver partition and portal vein ligation for staged hepatectomy (ALPPS), minimally invasive liver resection, ablation, liver transplantation

## Abstract

**Simple Summary:**

Colorectal cancer is a common cancer diagnosis, with many patients suffering from metastatic disease. Unfortunately, the prognosis for these patients remains grim, but there is a cohort in whom surgery remains a viable option, with the potential for a cure. The management of this cohort of patients is complex and requires the guidance of a dedicated hepatobiliary surgeon. There also remains a deep interplay between perioperative and operative decisions, which ultimately need tailoring to each individual patient. In this review, we aim to define these key perioperative and operative considerations in the contemporary management of colorectal liver metastasis.

**Abstract:**

Colorectal cancer is the third most common cancer in the United States and the second most common cause of cancer-related death. Approximately 20–30% of patients will develop hepatic metastasis in the form of synchronous or metachronous disease. The treatment of colorectal liver metastasis (CRLM) has evolved into a multidisciplinary approach, with chemotherapy and a variety of locoregional treatments, such as ablation and portal vein embolization, playing a crucial role. However, resection remains a core tenet of management, serving as the gold standard for a curative-intent therapy. As such, the input of a dedicated hepatobiliary surgeon is paramount for appropriate patient selection and choice of surgical approach, as significant advances in the field have made management decisions extremely nuanced and complex. We herein aim to review the contemporary surgical management of colorectal liver metastasis with respect to both perioperative and operative considerations.

## 1. Introduction

Colorectal cancer (CRC) is the third most common malignancy in the world and is the second leading cause of cancer-related mortality [1]. Approximately 20–30% of patients will develop synchronous or metachronous colorectal liver metastases (CRLMs) [2,3]; by definition, this is considered to be stage IV disease according to the AJCC TNM staging system [4]. Historically, metastatic disease was a contraindication to surgery with an associated 5-year overall survival (OS) of less than 5% after systemic therapy [5,6,7]. Even now, survival for all comers with stage IV disease remains low, at under 8% [3]. However, this can be quite misleading, as the AJCC TNM staging system fails to accurately account for resectable metastatic disease, in which there is a potential for cure. Indeed, early pioneers of resection demonstrated improved survival as early as the 1970s [8], and subsequent studies in the 2000s demonstrated that 5-year OS following hepatectomy can approach 40% [9]. Modern series even quote 10-year survival rates on the order of 20% following multimodal therapy that includes resection [10]. With time, the pool of resectable patients has increased significantly due to emerging surgical techniques and adjunct procedures [11,12], thereby offering more patients the potential for durable survival. In contemporary practice, it is recognized that a multidisciplinary approach to the management of cancer potentiates survival [13,14,15], and surgery remains a core tenet of that management as a potentially curative therapy [10].

### History of Liver Resection and the Goals of Resection

Liver surgery has made tremendous progress over the past several decades. While initial reports date back to the 19th century [16], the first detailed reports of resection for CRLM emerged in the 1940s [17]. This initial era of hepatectomy was characterized by poor postoperative mortality approaching 20% in some series [18]. However, as our anatomic understanding deepened [19], better hemorrhage control was employed through inflow occlusion [20], and low central venous-pressure anesthesia techniques were developed [21], perioperative mortality from hepatectomy decreased to less than 5% at specialized hepatobiliary centers [22], with most modern series quoting rates of 2–3% [23,24]. The goals of resection for those early pioneers are much the same as they are today, namely the eradication of all disease while maintaining sufficient future liver remnant (FLR) volume, perfusion, and biliary drainage to ensure adequate postoperative hepatic function. However, the emphasis has now shifted from what is technically resectable to appropriate patient selection according to disease biology. While many of the previous dogmatic teachings in liver surgery have been challenged, the contributions made by those early visionaries paved the way for our current understanding of this complex disease. Herein, we review the contemporary surgical management of CRLM with respect to both perioperative and operative considerations.

## 2. Perioperative Considerations

### 2.1. The Role of Early Surgical Evaluation

As the definition of resectability continues to change, the importance of obtaining an early surgical evaluation by a dedicated hepatic surgeon has become paramount. The number of patients that are potentially resectable has increased considerably from recent surgical advances [11,12] but the exact definition of resectability varies, even amongst a dedicated HPB faculty [25]. Considering that medical oncologists are often the primary point of contact for patients, especially in the metastatic setting, several patients with CRLM may incorrectly be deemed ineligible for curative intent therapy. In a recent article by Krell et al., a large portion of medical oncologists in Michigan noted factors such as the presence of bilobar disease, number of lesions, or tumor diameter size as contraindications to surgery, despite a lack of supporting evidence [26]. Further discrepancies between medical oncologists and hepatic surgeons regarding assessment of resectability have been well documented in the literature [27,28]. In fact, up to 20% of patients who are treated with palliative systemic therapy may be eligible for curative-intent locoregional therapy [14]. Thus, the underutilization of surgery and other local therapies may be excluding patients from otherwise potentially life-prolonging treatment [29]. These patients should be addressed in a multidisciplinary meeting that involves surgical input since cancer care in this setting has been shown to change management decisions in up to one-third of patients with GI malignancies [30]; for CRLM, this can mean providing a chance for cure.

### 2.2. Patient Selection and Factors Influencing Outcome

Patient selection for hepatectomy has changed with time. As previously mentioned, the early era of hepatectomy was characterized by high perioperative mortality [18], and patient selection was centered on technical factors. While anatomic factors remain the main determinants of resectability, as perioperative outcomes improved, it became clear that other factors, like tumor biology and genomics, were influencing outcomes as well; these factors should be assessed and taken into consideration when counseling patients regarding their prognosis. In Table 1 below, we describe some common anatomic factors of resectability, as well as clinicopathologic and genomic factors that impact survival.

#### 2.2.1. Anatomic Factors

Strictly speaking, the main anatomic factor for resection is maintaining an adequate future liver remnant (FLR) with complete resection of all disease; feasibility for resection has traditionally been described as preserving two contiguous segments with adequate vascular inflow, outflow, and biliary drainage [31].

#### 2.2.2. Tumor Biology Factors

Several scoring systems using clinicopathologic criteria have been published to predict tumor biology [32,33,34]. The most frequently used in clinical practice is the Clinical Risk Score (CRS), otherwise known as the Fong CRS [34]. After retrospectively reviewing 1001 hepatectomies for CRLM at a single institution, the authors identified five factors that were associated with long-term outcomes: node-positive primary, disease-free interval (DFI) from primary to metastasis diagnosis < 12 months, number of tumors > 1, largest hepatic tumor diameter > 5 cm, and carcinoembryonic antigen (CEA) > 200 ng/mL. Each factor was assigned a point value of 1, with increasing scores portending a worse disease-free survival (DFS). While initially designed as a method to select patients for clinical trial inclusion, the score has been widely used as a surrogate for disease biology.

**Table 1 cancers-16-00941-t001:** Anatomic factors of resectability and clinicopathologic and genomic factors impacting survival.

Anatomic	Clinicopathologic	Genomic
Tumor relation to vessels and bile ducts [31] Maintaining two contiguous segments [31] Vascular inflow, outflow, and biliary drainage [31] Adequate FLR [35,36] Ability to achieve an R0 resection	Disease-free interval [34] Primary tumor nodal status [34] Primary tumor sidedness [37,38] Number of lesions [34] Diameter of maximum lesion [34] CEA level [34]	KRAS [39,40,41] BRAF [42] MMR [43] SMAD-4 [44] FBXW7 [45] TP53-RAS [46,47]

Abbreviations: FLR—future liver remnant; CEA—carcinoembryonic antigen; KRAS—Kirsten rat sarcoma viral oncogene homolog; BRAF—V-Raf Murine Sarcoma Viral Oncogene Homolog B; MMR—mismatch repair; SMAD-4—mothers against decapentaplegic homolog 4; FBXW7—F-Box and WD Repeat Domain Containing 7; TP53—tumor protein p53.

#### 2.2.3. Genomic Factors

As next-generation sequencing (NGS) has become more widespread, several different genes and genomic pathways have been implicated in CRLM. Of these, one of the most widely studied alterations involves the RAS family of proto-oncogenes [48]. RAS mutations have been associated with a deleterious effect on both OS and progression-free survival (PFS), particularly in the metastatic setting [49]. In surgically resected patients, Kawaguchi et al., found that mutations in the RTK-RAS pathway were associated with worse OS compared to wildtype tumors [50]; however, this study grouped mutations involving KRAS and BRAF together. Vauthey et al., found that *mt*RAS specifically conferred worse 3-year OS (52.2% vs. 81%, *p* = 0.002) and RFS (13.5% vs. 33.5%, *p* = 0.001) vs. *wt*RAS in a resected cohort [41]. Margonis et al., further specified that it was codon 12 alterations that conferred significantly worse 5-year OS than *wt*RAS patients (34.4% vs. 46.9%, *p* < 0.05) and that the specific mutations responsible were G12V (HR 1.78, 95% CI 1.00–3.17; *p* = 0.05) and G12S (HR, 3.33; 95% CI, 1.22–9.10; *p* = 0.02) [39]. In another article, Margonis et al., compared outcomes of *mut*BRAF/*wt*RAS patients against *wt*BRAF/*wt*RAS; they found that *mut*BRAF—in particular, the V600E mutation—was dramatically associated with a worse OS (HR 2.39, 95% CI 1.53–3.72; *p* < 0.001) and DFS (HR 2.04, 95% CI 1.30–3.20; *p* = 0.002); they concluded that BRAF V600E was the strongest negative predictive factor for survival in resectable CRLM [42]. Other implicated genes in CRLM include TP53-RAS co-mutation [47], FBXW7 [45], and SMAD4 [44], among others. These data provide important prognostic information when counseling patients but also raise the possibility of tailored precision medicine, as is evident by the notion of mutation-specific postoperative surveillance [51].

Previous models for predicting outcomes were based on clinicopathologic factors, but it is evident that genomics plays a significant role in tumor biology. Authors hypothesized that incorporating both clinicopathologic factors and genomics may improve our ability to predict tumor biology. One such model is the RAS Mutation Clinical Risk Score [52], which is an update of the original CRS proposed by Fong et al. [34]. This modified CRS (*m*CRS) replaces disease-free interval, number of lesions, and CEA level from the original Fong CRS with RAS mutational status. The new *m*CRS was found to outperform the traditional Fong Score in a discovery and international multicenter validation cohort [52]. The two risk scores are depicted below in Table 2. An additional limitation that should be noted is that the original CRS was developed in an era where effective chemotherapy was still not readily available. As such, almost none of the patients included in that study received systemic therapy; progression in the setting of neoadjuvant therapy should be considered a surrogate for aggressive disease biology.

While clinical risk scores and other prognostic factors like genomics may help to potentially inform clinicians of a patient’s underlying tumor biology, they should not solely serve to exclude patients from surgery, as there are patients who fall into the worse risk groups that have had durable survival following hepatectomy.

### 2.3. Operative Sequencing for Synchronous Disease

The timing of a hepatectomy for synchronous CRLM can vary depending on a multitude of factors, including tumor burden, primary tumor location, and complexity of resection for example. The three widely accepted approaches include primary-first (PF), liver-first (LF), and simultaneous resection (SR). PF, which is sometimes referred to as the classical or traditional approach, involves resection of the primary cancer, followed by the liver disease later. LF reverses the sequence, with hepatectomy preceding primary tumor resection. Finally, in SR, both the primary and hepatic disease are resected under one general anesthetic. It should be noted, however, that regardless of which site of disease is resected first, most patients should receive preoperative chemotherapy prior to any surgical intervention.

#### 2.3.1. The Primary-First Approach

In general, the standard approach is PF, and this is particularly true in situations where the primary tumor is symptomatic or if there is an immediate indication for bowel surgery, such as uncontrollable hemorrhage, perforation, or obstruction. In these specific scenarios, an upfront resection of the primary may be considered in lieu of providing systemic therapy. Barring these acute scenarios, however, systemic therapy should be provided prior to resection in most cases. Additionally, there are potential risks and benefits to each of the other approaches that should be considered on a case-by-case basis.

#### 2.3.2. The Liver-First Approach

The LF approach was initially conceived to avoid issues associated with primary tumor resection that may result in a delay of chemotherapy administration, particularly in borderline resectable metastases [53]. In the first paper describing this approach, Mentha et al., examined 20 patients between 1999 and 2005 with synchronous CRLM. The patients underwent high-dose chemotherapy, using 5-FU/leucovorin or capecitabine, in addition to oxaliplatin and irinotecan, followed by curative intent hepatectomy and then resection of the primary lesion. Eleven patients had rectal primaries, while the remaining patients had colonic primaries; the median number of lesions was five, with a median size of 5.5 cm; and fourteen patients had bilobar disease. All patients had a Fong CRS of at least 3, while four patients each had a score of 4 or 5. In total, 4 patients developed disease progression while on chemotherapy, while the remaining 16 patients underwent resection, with 13 requiring major hepatectomy (three or more segments), 2 requiring portal vein embolization (PVE), and 4 requiring two-stage hepatectomy (TSH). The actuarial survival at 4 years for the patients who underwent resection after neoadjuvant therapy (16/20) was 61% [54]. In a subsequent, large-scale study between 2000 and 2010, using the LiverMetSurvey, a prospective international registry of patients undergoing CRLM surgery, Andres et al., compared outcomes of 729 patients undergoing the classical approach with 58 patients undergoing the LF approach. The groups were similar with respect to hepatic disease burden, but there were more rectal primaries in the LF arm and more clinically node-positive CRC primaries in the classical group. This study found that there was no difference in OS or RFS between the two approaches, thereby validating that the LF approach was a safe treatment algorithm [55].

This approach is particularly attractive for patients with rectal primaries. First, low anterior resection (LAR) and abdominoperineal resection (APR) are high-risk colorectal procedures that carry significant morbidity [56]. Additionally, even in the absence of immediate postoperative issues like anastomotic leaks, which occur in up to 3% of patients [57], other postoperative issues, like low anterior resection syndrome [58] or stoma issues, can complicate recovery. Since liver disease is the most prognostic site for metastatic CRC [59], any prolonged break from systemic therapy may result in progression to unresectability and thereby significantly worsen survival [60]. Finally, since contemporary management of advanced rectal tumors calls for a short course of neoadjuvant chemoradiation therapy with a subsequent treatment break to assess response, a natural window of opportunity presents itself for hepatectomy [61]. In this way, the LF approach also allows for patients to be eligible for watch-and-wait in the setting of a complete clinical response for the primary, thereby completely avoiding primary resection morbidity; note, however, that the watch-and-wait method was studied in localized rectal tumors without metastasis [62]. Nonetheless, the theoretical benefit does exist, as the recent literature has noted a complete pathologic response in the primary rectal tumor in 6% of patients who underwent the LF approach [53]. Indeed, recent consensus recommendations have recognized the LF approach as a viable treatment strategy, particularly for rectal primaries or in patients with borderline resectable hepatic disease [63]. There are some notable issues with the LF approach that should be highlighted, however. A notable percentage of patients who undergo the LF approach do not proceed to primary tumor resection, which is associated with a significant decrease in OS; in a recent article by Maki et al., the most common cause of failure to resect the primary in the LF approach was disease progression following hepatectomy (76%) [53].

#### 2.3.3. The Simultaneous Approach

For synchronous CRLM, the SR approach has been studied extensively. The proposed benefits of this approach include avoiding the morbidity and complications associated with performing a second operation, such as the inherent risks of anesthesia, need for a second hospital stay, potential for increased healthcare costs, and periprocedural pain, for example [64]. Additionally, this approach has the added benefit of addressing the liver disease immediately, thereby avoiding issues where disease progression can subsequently exclude hepatectomy. However, opponents of this approach have raised concerns regarding the safety of performing two potentially major operations concurrently [65]. To address these concerns, Tsilimigras et al., performed a propensity-matched analysis of approximately 1100 patients from a multi-institutional database of patients who underwent SR versus staged resection; in this study, LF and PF were grouped together. The authors noted a statistically significant increase in severe complications, defined as Clavien–Dindo grade III or greater, in the SR vs. staged-resection groups (16.9% vs. 7.0%; *p* = 0.002) but with comparable 90-day mortality (3.5% vs. 1.0%; *p* = 0.09). They did, however, note that severe complications in the SR group decreased over the study period (2008: 50% vs. 2018: 11.1%; *p* = 0.02). They then stratified their SR cohort into four risk groups depending on the risk of the associated hepatectomy and colectomy. They found that the risk of significant comorbidity and mortality increased significantly as the complexity of the operations increased, starting with low-risk hepatectomy/low-risk colectomy and moving toward high-risk hepatectomy/high-risk colectomy. There was no significant difference in survival between the SR and staged-resection groups in this study [66]. A recent propensity score-matched analysis of 590 patients also found that SR was associated with an increased risk of postoperative complications, but only in patients with a high Tumor Burden Score (OR 7.81, 95% CI 1.84–44.8, *p* < 0.01); in those with low or medium TBS, there was no increased risk of complications [67,68]. These survival data are in line with the previously published literature, including a network meta-analysis and the only randomized clinical trial on the subject [69,70]. Consensus guidelines recommend that a simultaneous approach can be undertaken when the hepatectomy and colectomy are both of low complexity [63,71].

In conclusion, there is merit to each surgical approach, and the decision to pursue one over the other requires an individualized approach to each patient, bearing in mind that there has been no difference in survival shown between them [64,69,72].

### 2.4. Chemotherapy Sequencing

Chemotherapy for CRLM has been employed in distinct clinical scenarios. In the setting of initially resectable or borderline resectable disease (defined as technically resectable but with a threatened R0 margin [73]), chemotherapy can be employed in the preoperative setting. When given preoperatively, chemotherapy affords the ability to treat radiographically occult disease while avoiding issues associated with adherence to systemic therapy postoperatively; it also tests tumor biology to assess for progression on chemotherapy, thereby providing prognostic information and potentially avoiding non-beneficial surgery. In the setting of unresectable CRLM, systemic treatment can be given as induction therapy in the hopes of converting the patient to resectable status. Alternatively, patients may also receive chemotherapy following resection as an adjuvant treatment. Giving hepatotoxic chemotherapeutic agents in this setting provides the benefit of avoiding chemotherapy-induced liver injury preoperatively, which can negatively impact postoperative outcomes [74]. There have been several trials attempting to elucidate the optimal sequence strategy, as we explore below.

#### 2.4.1. Adjuvant Therapy

Several randomized trials were carried out to assess the efficacy of various adjuvant chemotherapy regimens following hepatectomy, but they have been marred with poor patient accrual or issues with treatment compliance. An earlier trial comparing the efficacy of adjuvant 5-fluorouracil and leucovorin (5-FU/LV) to surgery alone demonstrated an improvement in DFS but no significant change in OS between the two groups [75]; similar results were demonstrated in a subsequent trial in which there was a tendency toward improved DFS but without a change in OS [76]. Both trials were stopped prematurely due to poor patient accrual. Mitry et al., through a pooled analysis of these two trials, found that there was a tendency toward improved median DFS and OS of 27.9 vs. 18.8 (HR = 1.32, 95% CI 1.00–1.76, *p* = 0.058) and 62.2 vs. 47.3 months, respectively (HR = 1.32, 95% CI 0.95–1.82, *p* = 0.095). This analysis demonstrated poor rates of postoperative treatment compliance (67%) [77]. To improve patient accrual and compliance, another trial examining adjuvant oral uracil–tegafur and leucovorin (UFT/LV) versus surgery alone was conducted with more relaxed inclusion criteria [78]. The addition of adjuvant UFT/LV improved the 3-year DFS (38.6% vs. 32.3%, *p* = 0.003) but without a change in OS compared to surgery alone. Other studies in the unresectable setting demonstrated an improvement in survival with the addition of irinotecan to 5-FU-based regimens, but this was not seen in another trial for resectable patients [79].

The most recent trial examining adjuvant therapy for CRLM is the JCOG0603 phase II/III randomized control trial, in which 300 patients between 2007 and 2019 were randomized to hepatectomy alone (149) versus hepatectomy plus adjuvant mFOLFOX6 (151); the primary endpoint of the trial was DFS. Using an intention-to-treat analysis with a median follow-up time of 59.2 months, the 5-year DFS was 38.7% (95% CI 30.4–46.8) for hepatectomy alone compared with 49.8% (41.0–58.0) for chemotherapy (HR 0.67; 95% CI 0.50–0.92; *p* = 0.006). However, the 5-year OS was 83.1% (74.9–88.9) and 71.2% (61.7–78.8) for the hepatectomy and chemotherapy arms, respectively (HR, 1.25; 95% CI 0.78–2.00; *p* = 0.42). Upon a subgroup analysis for DFS, patients with no prior chemotherapy exposure and those who had joined the trial after the second period of phase II benefited the most. This may be due to better adherence to systemic therapy, as the number of patients who completed the planned 12 courses of adjuvant therapy increased after protocol amendment following the first period of phase II from 36% at that time to 65% in phase III. It should be noted that the trial mostly comprised patients with a relatively low burden of hepatic disease (90% of patients had ≤3 lesions, while 86% of patients had a maximum tumor size of <5 cm). This trial demonstrated that there was a benefit of adjuvant therapy with respect to DFS but not to OS, which the authors noted may have been due to the adverse events experienced by patients that were accrued during the first period of phase II or the relative difficulty of identifying liver-only recurrence in the setting of chemotherapy (specifically oxaliplatin)-induced steatosis/steatohepatitis in the chemotherapy arm [80]. This trial was referenced in the most recent American Society of Clinical Oncology (ASCO) guidelines regarding the treatment of CRLM [81].

In summary, while the prospective randomized evidence has demonstrated a benefit for DFS for adjuvant therapy, a statistically significant benefit for OS has yet to be established.

#### 2.4.2. Perioperative Therapy

Perioperative chemotherapy has also been considered. Two major clinical trials to date have examined the use of perioperative chemotherapy for resectable CRLM: the EPOC and New EPOC trials [82,83,84,85]. In the EPOC trial, 364 patients were randomized to perioperative chemotherapy, with six cycles of FOLFOX4 before and after surgery versus surgery alone; the primary endpoint was PFS. With a median follow-up of 3.9 years and in all randomly assigned patients, the 3-year PFS was 28.1% vs. 35.4% (*p* = NS) in the surgery and perioperative chemotherapy arms, respectively, with an associated PFS HR for chemotherapy of 0.79 (*p* = 0.058). An analysis of eligible-only patients (11 patients excluded in each arm after randomization) demonstrated a 3-year PFS of 28.1% vs. 36.2% in the surgery versus perioperative chemotherapy arms with an associated HR of 0.77 (95% CI 0.60–1.00, *p* = 0.041). When considering only resected patients (approximately 150 in each arm), the 3-year PFS was 33.2% vs. 42.4% in the surgery and chemotherapy arms, respectively, with an HR of 0.73 (95% CI 0.55–0.97, *p* = 0.025) [82]. An updated analysis of this trial with a median follow-up of 8.5 years demonstrated that there was no difference in OS between the control and experimental arms but that the previously seen difference in PFS endured [83].

The New EPOC trial compared the effectiveness of perioperative standard chemotherapy (via mFOLFOX-6 or XELOX) with or without cetuximab, an anti-EGFR monoclonal antibody; this study was designed in response to other trials that demonstrated a survival advantage for anti-EGFR targeted therapy in *wt*KRAS patients with unresectable mCRC [86]. In total, 272 *wt*KRAS patients were randomized 1:1 to receive perioperative chemotherapy with or without cetuximab; most patients received mFOLFOX as the chemotherapy backbone (67% and 68% in both study arms). With a median follow-up of 20.7 months, the trial was terminated early due to futility: median DFS was 14.1 months in the chemotherapy plus cetuximab group and 20.5 months in the chemotherapy-alone group (HR 1.48, 95% CI 1.04–2.12, *p* = 0.03) [84]. A long-term update of the trial with a median follow-up of 66.7 months demonstrated a median OS of 81.0 vs. 55.4 months in the control and cetuximab arms, respectively (HR 1.45, 95% CI 1.02–2.05, *p* = 0.036) [85]. As such, the use of cetuximab in the perioperative setting was abandoned in favor of standard chemotherapy, possibly with bevacizumab.

The optimal treatment sequence for chemotherapy is still debated, but most societal recommendations still include perioperative chemotherapy, especially in the setting of synchronous disease [63].

### 2.5. Implications of Chemotherapy-Associated Hepatotoxicity

Should systemic therapy be administered prior to hepatectomy, careful consideration should be given to both the duration of therapy and the regimen employed, as these may impact the postoperative outcomes.

#### 2.5.1. Specific Regimens and Associated Toxicities

Prior studies have demonstrated that specific patterns of hepatic parenchymal injury, such as steatosis, steatohepatitis, and sinusoidal injury, are associated with worse postoperative outcomes. In a previous meta-analysis by de Meijer et al., examining 1000 patients undergoing major hepatectomy (defined as >3 Couinaud segments), the presence of hepatic steatosis was associated with an increased risk of postoperative morbidity (<30% steatosis: RR 1.53 [95% CI 1.27–1.85] | ≥30% steatosis: RR 2.01 [95% CI 1.66–2.44], both *p* < 0.001), as well as postoperative mortality (RR 2.79 95% CI 1.27–1.85, *p* < 0.001), compared to patients without steatosis [87]. A subsequent meta-analysis by Robinson et al., attempted to elucidate the specific causative regimens for chemotherapy-associated hepatotoxicity [88]. The authors found no association between oxaliplatin-based or irinotecan-based therapy and hepatic steatosis, but they did note significant heterogeneity in the studies examining irinotecan-based treatments [88]. This was likely related to considerable interobserver variability amongst pathologists [89,90] or varying degrees of steatosis present in patients prior to starting therapy, thus confounding the results [88]; indeed, some studies have noted an association between steatosis and BMI rather than chemotherapy [91]. Steatohepatitis, which is defined as the presence of hepatic steatosis with inflammatory infiltrate, is strongly associated with postoperative morbidity and mortality [74]. Robinson et al., demonstrated that irinotecan-based chemotherapy regimens were associated with an increased risk of steatohepatitis compared to chemotherapy-naive patients (relative risk 3.45 95% CI 1.12–10.62; *p* = 0.03) [88], and this may persist even 9 months beyond cessation of chemotherapy [92]. The meta-analysis found no association between oxaliplatin-based therapy and steatohepatitis [88]. Sinusoidal injury, which is characterized by sinusoidal obstruction with associated parenchymal atrophy, is also associated with worse postoperative morbidity [93]. Sinusoidal injury is associated with oxaliplatin-based regimens, with an RR of 4.36 (95% CI 1.36–13.97; *p* = 0.01) compared to chemotherapy-naive patients [88]; there is no such association with irinotecan-based therapies. It should be noted that sinusoidal injury may reverse with time [92].

#### 2.5.2. Duration of Therapy

Developing chemotherapy-associated hepatotoxicity is also based on the duration of therapy. Chen et al., aimed to define the optimal number of preoperative chemotherapy cycles after the propensity-score matching of 129 patients who all received neoadjuvant chemotherapy; after dividing the cohort into <5 cycle versus ≥5 cycle groups, they found that the ≥5 cycles group was more likely to develop hepatotoxicity (87.8% vs. 65.0%, *p* = 0.004) and that administration of ≥5 cycles was independently associated with a reduced PFS (HR = 2.26, 95% CI 1.28–4.01; *p* = 0.005) and OS (HR = 2.81, 95% CI 1.36–5.82; *p* = 0.005) [94]. Other groups have concluded that there is no significant increase in tumor response to chemotherapy beyond 4 months of therapy [95]. The EPOC trial included the administration of 6 weeks of preoperative therapy [82] but did not include information regarding the prevalence of histologically confirmed chemotherapy-associated liver injury from resected specimens. In this regard, the optimal duration of therapy may also vary depending on whether patients received adjuvant therapy following primary tumor resection, and so it is not clearly defined.

#### 2.5.3. Protection against Toxicity

Interestingly, bevacizumab was initially studied in the metastatic setting and was found to improve survival [96], but subsequent studies also found that it had protective effects against chemotherapy-associated hepatotoxicity [23]. This was initially reported in 2007 by Ribero et al., who examined 105 consecutive patients treated with 5-FU and oxaliplatin (5-FU/OX), of which 62 received bevacizumab. They found that bevacizumab not only improved pathologic response rates (32.9 vs. 45.3% residual viable tumor cells; *p* < 0.02) but also reduced the degree of sinusoidal injury (any grade of injury: 27.4% vs. 53.5%; moderate or severe grade of injury: 8.1% vs. 27.9%; both *p* < 0.01) [23]. The findings of reduced sinusoidal injury were later confirmed in separate studies by Rubbia-Brandt et al. [97] and Klinger et al. [98].

### 2.6. Disappearing Liver Metastasis

As chemotherapy is increasingly utilized in the management of CRLM, a unique and difficult clinical scenario regarding the management of disappearing liver metastasis (DLMs) has arisen. DLMs are lesions that were present on imaging prior to the initiation of chemotherapy but then demonstrated a radiographic complete response on interval imaging. These lesions pose several clinical challenges that we discuss further.

#### 2.6.1. Risk Factors for Developing DLMs

Several prior series have attempted to identify particular risk factors for the development of DLMs. In a study consisting of 40 patients who had 126 DLMs, van Vledder et al., found that >3 metastasis prior to chemotherapy (OR 13.1; *p* < 0.001) and the number of preoperative chemotherapy cycles (OR 1.18; *p* = 0.03) were independently associated with the development of DLMs [99]. In another series, Tani et al., found that patients with DLMs were more likely to have a smaller minimum size of their lesions and were more likely to be treated with oxaliplatin-based chemotherapy [100]. The risk factor of smaller lesion size has been identified in other series as well, as has the presence of synchronous disease (OR 11.25; *p* = 0.015) [101]. Taken together, these factors may help identify which patients are at highest risk of developing DLMs.

#### 2.6.2. Imaging Studies of CRLM

Contrast-enhanced CT (CE-CT) serves as the imaging modality of choice during the initial staging workup of CRC. However, in a systematic review of patients without prior therapy, Niekel et al., found that CE-CT had a per-patient and per-lesion sensitivity in regard to detecting CRLMs of 83.6% and 74.4%, respectively [102]. This article and many others have instead suggested that liver MRI be the first-line imaging modality for assessing CRLMs. However, given the lack of widespread availability of MRI and its relative cost compared to CT, it is not typically employed during the initial staging of CRC.

#### 2.6.3. Contrast-Enhanced CT

In the setting of neoadjuvant chemotherapy, however, previous studies have found that the sensitivity of CE-CT diminishes to as low as 65% [103], a finding that was also seen in a recent meta-analysis [104]. This is thought to occur because of chemotherapy-associated liver injury in which the ensuing steatosis, steatohepatitis, or sinusoidal obstruction syndrome may result in a decreased attenuation difference between normal hepatic parenchyma and the CRLM [104,105,106]. In fact, another meta-analysis consisting of patients exclusively treated with neoadjuvant chemotherapy found that the pooled sensitivity of CE-CT was 69.9% compared to 80.5% in chemo-naive patients [105]. This same study found that the pooled sensitivity of MRI was 85.7% in the setting of neoadjuvant chemotherapy [105].

Despite this, CE-CT remains the preferred imaging modality during the initial staging workup of CRC because of its cost-effectiveness, ability to assess the entire abdomen, and widespread availability. In the setting of CRLMs identified by CT, however, there may be a role for subsequent MRI, particularly in the setting of neoadjuvant chemotherapy.

#### 2.6.4. Liver MRI

According to the previously published literature, approximately 11–37% of patients treated with neoadjuvant chemotherapy may develop one or more DLMs [107,108]. However, because the sensitivity of CE-CT decreases as a result of neoadjuvant chemotherapy, other imaging modalities, such as MRI enhanced with gadoxetate disodium or additional diffuse-weighted images (DWIs), as well as PET/CT, have been investigated in evaluating DLMs.

In a meta-analysis and systematic review aiming to determine the diagnostic accuracy of various imaging modalities for CRLMs, Choi et al., found that the pooled sensitivity for CT, gadoxetate-enhanced MRI, and PET/CT was 82.1% (74.0–88.1%), 93.1% (88.4–96.0%), and 74.1% (62.1–83.3%), respectively. The pooled specificity for CT, MRI, and PET/CT was 73.5% (53.7–86.9%), 87.3% (77.5–93.2%), and 93.9% (83.9–97.8%), respectively. MRI was significantly more sensitive than CT (*p* < 0.001) in this study [104]. In another study of 29 patients with DLMs, Sturesson et al., found that the preoperative but post-chemotherapy sensitivity of CT and gadoxetate-enhanced MRI was 53% and 36%, respectively, with no statistical difference observed between the imaging modalities (*p* = 0.312) [109]. Macera et al., examined the effects of adding DWI to gadoxetate-MRI in 32 patients with 144 CRLMs. The authors found the sensitivity of gadoxetate-MRI to be 73.6% overall, but it was 94.6% and 100% for lesions of 10–20 mm and >20 mm, respectively. However, for sub-centimeter lesions, the sensitivity of gadoxetate-MRI plummeted to 52%. In comparison, DWI + gadoxetate-MRI had a similarly high sensitivity for lesions 10–20 mm and >20 mm (both 100%) but had a much higher sensitivity, i.e., 81.3%, for sub-centimeter lesions [110]. As we noted earlier, it is generally the smaller lesions that are at particular risk of becoming DLMs, and so gadoxetate-MRI with DWI may serve as a particularly valuable imaging modality in these patients.

#### 2.6.5. The Role of Intraoperative Imaging

Since no imaging modality has a sensitivity of 100%, a thorough intraoperative assessment using full liver mobilization, inspection, palpation, and IOUS is typically employed to assess for lesions. In a retrospective review, Ferrero et al., identified 33 patients with 67 DLMs that underwent hepatectomy. Twenty-six (79%) patients had preoperative gadoxetate-MRI. Of the 67 DLMs, 45 (67%) were discovered at the time of laparotomy, using IOUS. The median size of the lesions detected on IOUS was 6 mm, and the factors independently associated with discovery on IOUS included moderate-to-severe hepatic steatosis (*p* = 0.016) and subglissonian location of the nodules (*p* = 0.019) [108].

CE-IOUS has also emerged as an adjunct in the detection of DLMs. Arita et al., retrospectively analyzed 72 patients who had undergone preoperative gadoxetate-MRI and CE-CT in addition to IOUS and CE-IOUS. Of these 72 patients, 11 developed 32 DLMs. Upon laparotomy, four DLMs were initially identified using IOUS. Subsequent CE-IOUSs identified those four lesions and an additional 12 DLMs. The sensitivity of the true positive DLMs, which were defined as the lesions that were histologically confirmed as being CRLMs, for IOUS and CE-IOUS was 21% and 79%, respectively (*p* < 0.004) [111]. In a subsequent prospective nonrandomized trial, Arita et al., examined the effects of CE-IOUS in 100 consecutive patients with 242 liver lesions identified preoperatively, using CE-CT and gadoxetate-MRI. Patients subsequently underwent sequential IOUS, followed by CE-IOUS, and the diagnosis of each liver lesion was confirmed histologically either through resection or core-needle biopsy; if no lesion was identified on IOUS/CE-IOUS, then it was left in situ and was defined as a durable clinical response if no mass-occupying lesion was identified on surveillance imaging 6 months post-hepatectomy. Of the 242 lesions identified preoperatively, CE-IOUS identified 239, of which all were histologically confirmed as CRLMs; the remaining 3 lesions were benign (1 biopsied and 2 with no growth on imaging). IOUS identified 25 additional lesions not seen preoperatively; CE-IOUS identified 22/25 lesions, of which 21 were histologic CRLMs (95%). Of the three lesions seen on IOUS but not on CE-IOUS, one was resected and histologically benign, while the other two had durable clinical response. Essentially, CE-IOUS correctly identified three (75%) of the false positives of IOUS. CE-IOUS identified another 22 nodules that were not seen by IOUS, of which 17 were CRLMs. The authors noted a sensitivity of CE-IOUS of 99%, PPV of 98%, and accuracy of 97% for CE-IOUS [112].

#### 2.6.6. Factors Associated with Complete Pathologic Response

The ideal clinical scenario would involve knowing which lesions reflected a complete pathologic response so as to avoid unnecessary surgery. While no imaging modality is 100% accurate, several prior studies have attempted to identify characteristics associated with complete pathologic response. Auer et al., noted that use of hepatic arterial infusion (HAI) chemotherapy (OR 6.2; *p* = 0.02), lack of visibility of the DLM on an MRI (OR 4.7; *p* = 0.005), and normalization of serum CEA levels (OR 4.6; *p* = 0.006) were independently associated with a complete pathologic response [113]. In another study examining 168 DLMs in 43 patients, Kim et al., identified that reticular hypointensity of the liver parenchyma on hepatobiliary phase imaging was associated with in situ recurrence at 2 years postoperatively (35.9% vs. 4.9%; *p* = 0.006) [114]. In a systematic review assessing the ability of various imaging modalities to predict true pathologic responses for DLMs, Muaddi et al., found that the highest pooled NPV, which they defined as the proportion of true negative DLMs, was 0.79 for CE-IOUS and 0.73 for gadoxetate-MRI [115].

#### 2.6.7. Management Strategies and Outcomes

The management of DLMs is difficult and is partly based on the inability to accurately depict whether lesions have viable disease post-chemotherapy, which can be up to 89% in some series [109]. Consensus expert statements recommend that all sites of metastatic disease be resected when possible [116], but in patients with extensive disease, surgical resection may lead to significant parenchymal loss, even in the setting of the parenchymal-sparing approaches. Oba et al., identified 275 DLMs in 184 patients, of which 110 lesions were not detected by either preoperative gadoxetate-MRI or CE-IOUS. Of these 110 lesions, 68 were resected by “blind hepatectomy” or anatomic resections that were thought to include the lesions based on pre-chemotherapy imaging. Of the 68 lesions undergoing blind hepatectomy, 3 (4%) contained viable disease. Of the remaining 42 DLMs left in situ, 6 (14%) developed recurrence, with a median follow-up time of 27 months [117]. Similarly, Sturesson et al., also identified 20 DLMs that were not identified by either gadoxetate-MRI or CE-IOUS but underwent blind hepatectomy; none of the resected lesions had residual disease [109].

These findings point to an important treatment paradigm adopted by surgeons in which lesions not readily identified by both preoperative and intraoperative imaging are less likely to harbor residual disease [118]. In an international survey with 226 respondents, Melstrom et al., found that 63% of surgeons would wait a specific period of time off of chemotherapy in the setting of DLMs to assess the durability of response prior to proceeding with resection. In situations where the DLMs were not identified using IOUS, 48% of surgeons would elect for observation, while only 31% would resect the presumed site of disease if superficially located; of those that elected to observe, the majority (84%) stated that the lesion could be treated later if it were to progress [118].

Outcomes regarding lesions that are left in situ versus those that are resected have also been investigated. Notably, several series have demonstrated that patients who have all sites of metastatic disease resected based on pre-chemotherapy imaging have improved RFS compared to those in whom DLMs are left in situ [99]. However, it should be noted that, despite a decrease in intrahepatic RFS, van Vledder et al., found no difference in the 3-year OS (70.8% vs. 63.5%, *p* = 0.66) between patients who had DLMs resected versus those left in situ [99]. Interestingly, Kuhlmann et al., found that, with a median follow-up time of 30 months, the 3-year OS for patients with DLMs was 64.7% vs. 30.2% (*p* < 0.05) for those without DLMs [119]. This likely represents that the presence of DLMs serves as a surrogate for better chemotherapy response, which, in turn, leads to improved outcomes.

DLMs are a complex and difficult scenario that HPB surgeons will face. Having the appropriate preoperative imaging, as well as a thorough intraoperative assessment, may aid in the management of this challenging entity.

### 2.7. Hepatic Arterial Infusion Pump: Unresectable Disease and Conversion Therapy

#### 2.7.1. Conceptual Basis

CRLM is unresectable in up to 80% of patients, in whom the 5-year OS is very poor [120]. To treat these patients, hepatic arterial infusion pump (HAIP) chemotherapy was developed. Conceptually, HAIP chemotherapy selectively delivers chemotherapeutic agents into the hepatic arterial system, which provides the dominant blood supply to liver metastasis. Conversely, normal hepatic parenchyma receives dual supply from both the artery and portal system [121]. Thus, using a chemotherapeutic agent with a high rate of first-pass metabolism, such as floxuridine (FUDR), allows for the delivery of extremely concentrated doses of therapeutic agent, while simultaneously limiting systemic toxicity [122].

#### 2.7.2. Trial Data

HAIP for CRLM was studied in the unresectable setting and to convert patients to resectability. In a phase II prospective trial, 49 unresectable CRLM patients were treated with HAIP in conjunction with standard systemic therapy to assess for conversion to resection (CTR) and survival. In this cohort of heavily pretreated patients with a high tumor burden (median number of tumors 13 and CRS > 3 in 90%), CTR at a median of 6 months was 47% [123]. In an updated analysis with 23 additional patients and a median follow-up time of 81 months, the 5-year OS for the entire cohort was 36% and 51% in the patients who were chemotherapy-naive; CTR was 52%. The subgroup of patients who underwent resection within 1 year of treatment initiation had a 5-year OS of 63%, according to a prespecified landmark analysis [124]. A more recent study in 2021 highlights the important role that HAIP may play in patients with chemotherapy-refractory progression of disease [125]. O’Leary et al., examined 25 unresectable CRLM patients, all of whom were treated with FOLFOXIRI, bevacizumab, and anti-EGFR inhibitor when appropriate. These were patients with a high tumor burden (72% with >10 liver metastasis and 48% with extrahepatic disease) who were treated with FUDR pump therapy in addition to concomitant systemic therapy, most commonly with FOLFIRI (60%). The total disease control rate was 80%, as defined as 40% partial response and 40% stable response in the best RECIST-defined [126] hepatic response. The median OS for the entire cohort was 11 months versus 14 months in liver-limited disease and 8 months for those with concomitant extrahepatic disease (*p* = 0.06) [125]. In comparison to the otherwise marginal benefit achieved from regorafenib (6.4 vs. 5 months in placebo) [127] or TAS-102 (7.1 vs. 5.3 months in placebo) [128], the impact of FUDR-HAIP on survival in this setting was substantial.

#### 2.7.3. Biliary Sclerosis

There are, however, some complications associated with the HAIP chemotherapy, most notably biliary sclerosis (BS) [129]. In a contemporary analysis of 475 consecutive patients who received HAIP for either adjuvant therapy or unresectable CRLM, Ito et al., identified that BS occurred in approximately 5.5% of patients in the adjuvant setting and 2% of unresectable patients [130]. The authors noted that the risk factors for BS were postoperative infectious complications (50.0% vs. 14.8%, *p* = 0.002) and larger dose/cycle/weight of FUDR (2.6 vs. 2.0 mg/cycle/kg, *p* = 0.025); they also noted that the development of BS did not negatively impact survival in these patients (BS vs. non-BS: 61.0 months vs. 47.2 months, *p* = 0.316). Additionally, prior studies have demonstrated that biliary toxicity from HAIP can be limited by the concomitant administration of dexamethasone with FUDR [131] and that an increase in alkaline phosphatase (ALP) may be an early indicator of toxicity [132]. As such, frequent lab monitoring can be used for surveillance purposes, and dose adjustments or treatment breaks have been shown to limit BS development [133].

#### 2.7.4. Adoption

Despite the encouraging results, there are still some significant barriers to the widespread adoption of HAIP. Firstly, much of the prospective data surrounding its use come from a single pioneering institution in the technology, which can significantly limit its external validity. From a systems perspective, the technology requires expert personnel with specific training available only at a handful of academic centers. Finally, it also requires buy-in from surgeons, medical oncologists, interventional radiologists, and others in both providing the treatment but also managing complications that may arise. In recent years, however, there has been a renewed interest in HAIP, with more and more centers around the world adopting its use. As a result, the HAI Consortium Research Network (HCRN) was established in 2020 and seeks to conduct a phase III randomized trial comparing HAI with systemic chemotherapy versus systemic chemotherapy alone for unresectable CRLM [134].

## 3. Operative Considerations

### 3.1. Portal Vein Embolization, Liver Venous Depletion, and Functional Assessment

#### 3.1.1. Portal Vein Embolization History and Volumetric Requirements

The liver’s regenerative capabilities were noted in the surgical literature as early as the 1980s [135]. Takayasu et al., under the guidance of Dr. Makuuchi, noted ipsilateral lobar atrophy with contralateral hypertrophy in a patient with hilar cholangiocarcinoma invading into the portal vein [135]. Upon making this observation, the same group subsequently postulated that intentional diversion of portal flow can induce liver hypertrophy via PVE [136]. This served as a major step forward in the management of CRLM, particularly in situations where major hepatectomy was necessary for tumor clearance. In 2000, Azoulay et al., published a 10-year experience of PVE; they found that initially unresectable patients that underwent PVE and were subsequently resected had an OS similar to those who were resected upfront, without an increase in perioperative morbidity or mortality [137]. Their results were subsequently reproduced by other groups [138]. Notably, it became increasingly evident that underlying hepatic dysfunction was a critical determinant in the ideal FLR. In those without underlying disease, FLR can be as low as 20%. However, as the effects of chemotherapy-associated hepatotoxicity came to light, it became evident that an FLR of 30% was necessary in this setting [36], and in those with end-stage liver disease/cirrhosis, an FLR of 40% is considered the minimum [35].

#### 3.1.2. FLR Function Assessment

Despite an adequate FLR volume (FLR-V), there are still reports of patients who develop post-hepatectomy liver failure (PHLF), highlighting that the underlying function of the FLR may matter more than the FLR-V [139]. Methods of assessing FLR function (FLR-F) include hepatobiliary scintigraphy (HBS) or SPECT-CT [139,140]. De Graaf et al., examined 55 patients undergoing major hepatectomy who had both preoperative PVE and HBS. In their cohort, nine patients developed PHLF, and three of those patients had an adequate FLR-V. In light of this, they advocated for the use of HBS to assess FLR uptake function, where a cutoff of 2.69%/min/m^2^ can be used as a means to determine whether hepatectomy can be safely performed; this cutoff outperformed FLR-V in predicting PHLF with respect to sensitivity, specificity, positive predictive value, and negative predictive value [139]. However, given some limitations of the HBS scan itself, there may be some overestimation of the function of the left hemiliver and underestimation of the function of the right. Combining HBS with SPECT has been demonstrated to improve the accuracy of functional studies even further compared to HBS alone [140]. The kinetic growth rate of the FLR has also been proposed as a more accurate predictor of PHLF than FLR-V [141]. Shindoh et al., in an another article, demonstrated that a kinetic growth rate of 2.0% per week was a better predictor of overall and liver-specific postoperative morbidity, as well as mortality, when compared to standardized FLR and degree of hypertrophy (area under curve 0.83 [95% CI, 0.736–0.923], *p* < 0.002) [141].

#### 3.1.3. Liver Venous Depletion

In situations where patients require an even more dramatic increase in FLR-V, a newer technique called liver venous depletion (LVD) can be performed, which entails concurrent embolization of the ipsilateral portal vein, along with the corresponding hepatic vein [142]. Le Roy et al., compared 41 PVE and 31 LVD patients and found that LVD had an increased percentage of FLR ratio hypertrophy (51.2% vs. 31.9%, *p* = 0.018) and kinetic growth rate (19% vs. 8%, *p* = 0.026) when compared to PVE alone [142]. Indeed, Heil et al., noted that patients who underwent LVD had increased rates of resectability (90 versus 68%; *p* = 0.007) compared to PVE in a multicenter retrospective review [143]. An ongoing prospective clinical trial, the DRAGON trial, which compares LVD and PVE is underway.

#### 3.1.4. PVE Considerations

For all the benefits that PVE provides, there are still some drawbacks, namely in the form of accelerated post-procedural tumor growth [144,145]. Kokudo et al., compared 18 patients with preoperative PVE who underwent major hepatectomy compared to 29 patients who underwent resection alone. They found that, in patients undergoing PVE, there was a statistically significant increase in percent tumor volume and Ki-67 compared to the non-PVE group (46.6 vs. 35.4%, *p* = 0.013; and 16.2 vs. 13.7%, *p* = 0.014, respectively) [145]. Pamecha et al., similarly demonstrated that PVE resulted in an increased tumor growth rate compared to the controls (0.36 mL/day vs. 0.05 mL/day; *p* = 0.06) [144]. This highlights the issues associated with PVE, particularly in the setting of extensive bilobar disease, in which post-procedural tumor growth may result in progression to unresectability.

### 3.2. Resection Margin

#### 3.2.1. Historical Perspective of the 1 mm Margin

Historically, the recommended resection margin for CRLM was 1 cm [146,147,148,149]. At the time, this forced hepatic surgeons to balance the constraints of maintaining an adequate FLR with obtaining appropriate margins for an oncologically sound procedure. Proponents of narrowing the resection margin argued that it would increase the pool of patients eligible for curative-intent therapy. Pawlik et al., demonstrated that patients with a sub-centimeter margin had improved survival compared to those who had a positive resection margin, which they defined at the time as <1 mm [150]. Additionally, they found that, in their patient population, 47% of patients did not achieve a resection margin of 1 cm [150], which was consistent with other reports at the time [146,151]. As a result of this prior work, a formal recommendation for a 1 mm margin was made by the Expert Group on OncoSurgery management of LIver Metastases (EGOSLIM) [71]; however, even this cutoff was scrutinized. Sadot et al., retrospectively examined 2368 patients undergoing hepatectomy for CRLM at a single institution; they found that patients with a resection margin of ≥1 mm were found to have improved survival outcomes when compared to those with submillimeter margins. However, the patients with submillimeter margins still had improved survival compared to a microscopically involved margin [152]; similar results were seen in studies from other institutions as well [153].

In general, the currently accepted minimum margin is 1 mm but with the caveat that patients at risk of having a sub-millimeter margin should still be offered resection, as there appears to be an associated survival benefit. Still, there have been studies that questioned the value of even a microscopically involved margin. De Haas et al., demonstrated that there was no difference in survival between R1, defined as <1 mm, and R0 resections in 436 consecutive patients at their single institution; while this study was criticized for its methodology, it still paved the way for others to examine factors that may influence margin [154].

#### 3.2.2. Vascular R1 versus Parenchymal R1

The use of IOUS in PSS led to the distinction between R1*par*, in which the parenchymal margin is involved with tumor cells <1 mm from the transection edge, and R1*vasc*, in which a tumor is detached from a major vascular structure [155]. Vigano et al., found that patients who had an R1*vasc* margin had equivalent outcomes to R0 resection and that R1*par* patients fared significantly worse [155], thus suggesting that the vessel itself may serve as a barrier to tumor spread. It should be noted, however, that the quality of the evidence regarding R1*vasc* versus R1*par* is limited to retrospective studies and likely needs further investigation.

#### 3.2.3. The Effect of Margin Stratified by Other Prognostic Factors

As perioperative chemotherapy became more prevalent, understanding the interplay between margin and systemic therapy garnered interest. Ayez et al., examined the effect of neoadjuvant chemotherapy administration on outcomes stratified according to margin status [156]. They found that, in patients who did not receive neoadjuvant chemotherapy, an R1 resection, defined as <1 mm, portended a significantly worse OS and DFS compared to an R0 resection but had no effect in those who received neoadjuvant therapy [156]; similar results were demonstrated in subsequent studies as well [157]. Other authors quantified the association between margin and response to neoadjuvant therapy rather than just its administration. Andreou et al., found that patients with a >50% pathologic response to systemic therapy had no difference in OS between R0 and R1 resections (63 vs. 67%, respectively; *p* = 0.587), but R1 resection conferred significantly worse survival in patients, with a <50% response (46 vs. 0%, *p* = 0.002) [158]. Other groups found that adjuvant therapy was also associated with improved outcomes [157,159,160].

As NGS became more widespread, researchers also examined the association of margin status and genomics as well. Brudvik et al., found that the presence of *mt*KRAS was associated not only with an increased likelihood of having an R1 resection compared to *wt*KRAS (11.4% vs. 5.4% *p* = 0.007) but was also associated with worse OS (HR 1.629; *p* = 0.044); in this patient cohort, 86% of patients received neoadjuvant therapy [161]. A subsequent study by Margonis et al., found that, in patients with *wt*KRAS tumors, a resection margin of >1 mm was associated with improved OS, but in those with *mt*KRAS tumors, resection margin had no prognostic value in predicting survival [162]. Xu et al., then examined the effect of margin on survival according to both response to neoadjuvant therapy and the presence of *mt*KRAS [163]. They found that, in patients with “good” tumor biology, as defined by *wt*KRAS and a response to neoadjuvant chemotherapy, R1 resection had no effect on the 5-year OS compared to R0. However, when examining patients with “bad” tumor biology, as defined by the presence of *mt*KRAS or no response to neoadjuvant chemotherapy, R1 resection was associated with a significantly worse 5-year OS compared to R0. These data are depicted in Table 3 below [163].

Thus, while the goal of resection should always be to obtain a margin of at least 1 mm, other factors, such as response to systemic therapy or mutational status, may overshadow the prognostic value of a positive resection margin.

Next, we will cover a variety of surgical approaches that can be used to address both synchronous and metachronous disease. A sample flow diagram illustrating how these different approaches can be utilized is depicted in Figure 1 below.

### 3.3. The Two-Stage Hepatectomy: A Method to Address Bilobar Disease

#### 3.3.1. Description and Historical Perspectives

The TSH was formally described in 2000 by Adam et al., following a report from 1996 describing resection of initially unresectable disease following neoadjuvant chemotherapy [164,165]. The original description of the procedure involved resection of as many metastases as possible in the first stage, followed by an interval of time where the remaining liver could regenerate, followed by resection of the remaining disease in stage 2 [165]. This procedure was successful in 13 of the 16 patients initially described. Subsequent modifications to this technique involved employing methods to increase the FLR. Kianmanesh et al., described a modified technique in which the left lobe of the liver is “cleared” through a combination of wedge resections and ablations, followed by ligation of the right portal vein; a subsequent interval of 4–8 weeks was allowed for FLR growth, followed by resection of the remaining disease in the right liver [166]. Another slight modification of this technique was proposed by Jaeck et al., in which, instead of performing an operative portal vein ligation (PVL), percutaneous PVE was employed 2–5 weeks after the first stage, with completion of the hepatectomy 2–3 months after PVE. Jaeck et al., proposed that this modification may decrease the likelihood of patients developing PHLF after the first stage of the procedure [167]. Notably, they also described performing the first stage of the procedure with resection of the primary tumor [167].

#### 3.3.2. Oncologic Outcomes and Failure to Progress

Wicherts et al., described long-term follow-up data in 59 patients who underwent TSH. Failure to proceed to the 2nd stage occurred in approximately 30% of the cohort, almost all of which were due to disease progression in the interim between resections. Of those that completed both stages, the 5-year OS was 42% and is noted by the authors to be equivalent to patients offered upfront, one-stage hepatectomy. However, in those that failed to complete both stages (18), no patient survived past 19 months [168]. In a systematic review conducted by Lam et al., 459 patients undergoing TSH were evaluated, and 23% of patients failed to progress to the second stage of the procedure, with disease progression being the most common reason (88%) [169]. In those that did complete both stages, 5-year median OS was 42% [169]. A more recent, multi-institutional analysis across five major hepatobiliary centers in the US examined outcomes from 196 patients who completed both stages of the TSH. In those that complete both stages, predictors of worse 5-year OS included an R1 margin (55% vs. 10%, *p* < 0.032) and greater intraoperative blood loss [170]. The authors found that 47.4% of patients experienced morbidity after the 2nd operation, with 23.4% experiencing major morbidity, defined as Clavien–Dindo class III or greater [170].

These studies highlight some key aspects of TSH. Firstly, TSH is an oncologically sound approach to treating patients with extensive bilobar CRLM. However, there is significant morbidity associated with the procedure, particularly in the 2nd stage, as most resections (75.5% in the US multi-institutional analysis) are major hepatectomies involving ≥4 segments [170]. Finally, there remains a significant issue with patient selection for TSH, as approximately 20–30% of patients fail to progress to the 2nd stage of the operation, with the vast majority of them failing due to progression of disease [168,169]. In the cohort of patients who fail to complete the 2nd stage of the disease, the median OS is quoted to be as low as 16 months [169].

To minimize the number of patients who fail to complete the 2nd stage of the operation, the associating liver partition and portal vein ligation for staged hepatectomy (ALPPS) procedure was devised [171].

### 3.4. Associating Liver Partition and Portal Vein Ligation for Staged Hepatectomy

#### 3.4.1. Description and Early Perioperative Morbidity

ALPPS was the next entry in managing extensive bilobar CRLM. Like TSH, ALPPS was initially described as a two-staged procedure comprising right portal vein ligation with interlobar parenchymal transection during the 1st stage, followed by a short-time interval to allow for liver growth and then resection of the remaining disease [171]. Its main advantage over TSH is rapid growth of the FLR, such that optimal FLR-V could be achieved in as little as 9 days [172]; this, in turn, increased the number of patients who completed the 2nd stage to 100% [172]. These early results garnered a lot of enthusiasm, but opponents of this approach highlighted its significant periprocedural morbidity and mortality. The first major series of 202 patients demonstrated major morbidity, as defined as Clavien–Dindo class ≥3, and mortality of 27% and 9%, respectively [173]. Despite being an improvement over earlier ALPPS reports, where mortality approached 12% [172], this was still significantly higher than previously reported periprocedural mortality of 3% for the 2nd stage of TSH [169]. Subsequent efforts were centered on reducing perioperative morbidity and mortality through modification in techniques [174,175,176,177,178,179,180,181] and the adoption of a minimally invasive approach [182,183,184]. As a result, the morbidity of the procedure was transferred from the first to the second stage, parenchymal transection spared the hilum and was limited to 50–80% rather than complete transection, and a periprocedural PVE was employed instead of PVL; this new modification was termed the “Mini ALPPS” [181,184]. Later studies from the international ALPPS registry demonstrated that perioperative mortality had improved to 3.8% [175].

#### 3.4.2. Two-Stage Hepatectomy vs. ALPPS

While comparisons between ALPPS and TSH were performed using retrospective data [185], prospective data were lacking until the LIGRO trial [11]. This trial compared 97 patients, 48 undergoing ALPPS and 49 undergoing TSH, with the primary endpoint of identifying a difference in rates of completed resection between the two groups. In total, 92% of patients in the ALPPS arm compared to 57% in the TSH arm completed both stages of the procedure, and there was no difference in periprocedural morbidity or mortality between the two groups. However, some important considerations should be noted when interpreting the results of the study. Given the shorter duration between first and second stage for ALPPS compared to TSH, progression of disease is far less likely to hinder completion of both stages of the procedure since less time elapses for the disease to progress [169]. Also, a large percentage of patients in the TSH group failed to complete the 2nd stage due to inadequate FLR growth, which may be a reflection of inadequate PVE [186,187] rather than the procedure itself. The median OS for the TSH group was also low (26 months) compared to that of the previous literature and more in line with patients who only completed the 1st stage of TSH [188]. Thus, there may be an unknown confounder influencing OS in the TSH group that was not accounted for, such as genomic differences between the arms.

ALPPS likely does play a role in the management of extensive bilobar CRLM, but its role has yet to be fully elucidated; most hepatobiliary centers instead employ TSH.

### 3.5. Parenchymal-Sparing Surgery

Another strategy for maximizing FLR after extensive resections is to employ parenchymal-sparing surgery (PSS) techniques. This entails leaving a minimal amount of normal hepatic parenchyma around a resected tumor while maintaining an adequate margin, typically with the use of intraoperative US (IOUS). Employing this technique could reduce the number of major hepatectomies by up to 80%, and subsequent recurrences can be re-resected with excellent 5-year OS [189]. Torzilli et al., validated the use of IOUS and subsequently demonstrated that this technique could also be employed for lesions near the hepatocaval confluence, a location that would otherwise require a major hepatectomy with possible vascular reconstruction [190,191]. Gold et al., noted that, with the increasing adoption of this technique over an 11-year period, there were no differences in oncologic outcomes, but there was a decrease in major hepatectomies with associated decreases in perioperative mortality [192].

#### Post-Hepatectomy Recurrence following PSS

PSS also leaves the possibility of further local treatment in the setting of liver-only recurrence after initial resection. Studies have demonstrated that patients who are eligible for repeated local treatments after recurrence have equivalent survival to those who were treated initially with resection [193,194]. In a subgroup analysis of patients with liver-only recurrence following initial hepatectomy, Mise et al., demonstrated significantly improved 5-year OS from the date of the initial surgery (72.4% vs. 47.2%, *p* = 0.047) and from the date of recurrence (73.6% vs. 30.1%, *p* = 0.018) for PSS versus anatomic resection. For patients with extensive bilobar disease, an “extended one-stage hepatectomy” (E-OSH) employing extensive use of PSS principles has been described as an alternative to TSH [195]. Torzilli et al., compared patients with an extremely high tumor burden (median number of lesions 12, with at least 1 in contact with a major intrahepatic vessel) who underwent either E-OSH at one institution versus TSH at another. The authors found that patients who underwent E-OSH had a similar 5-year OS to those who completed both stages of the TSH (38.2 vs. 31.8%), with significantly less blood loss, overall morbidity, hospital stay, OR time, liver-specific morbidity, and need for blood transfusions, without differences in RFS or recurrence site. In an intention-to-treat analysis, patients undergoing E-OSH tended to have a better 5-year OS compared to patients for whom TSH was attempted (38.2 vs. 21.1%, *p* = 0.071) [195]. In comparison to ALPPS, E-OSH has also been shown to have similar OS (31.7 vs. 32.6 months) and DFS (10.6 vs. 7.8 months) but was noted to have significantly less morbidity (26.9% ALPPS vs. 7.7% E-OSH, *p* = 0.017) [196].

Given these data, the standard is to employ a PSS approach for hepatectomy while reserving major liver resections or TSH for patients in whom an R0 resection is threatened.

### 3.6. Minimally Invasive Liver Resection

#### 3.6.1. Laparoscopic Liver Resections

There have been two randomized clinical trials comparing open liver resection (OLR) and laparoscopic liver resection (LLR): the LapOpHuva [197] and OSLO-COMET [198] trials.

In the LapOpHuva trial, 193 patients were randomized to either OLR or LLR; the primary endpoint of the trial was 90-day morbidity, and the secondary outcomes were 90-day mortality, OR time, estimated blood loss (EBL), transfusion rate, Pringle time, length of stay, OS, and DFS. The two arms were well matched, with an equal proportion of patients with bilobar disease, disease in segments VI–VIII, and rates of major hepatectomy; of note, most patients had a small number of lesions (1–2) that were 3–4 cm. Postoperative morbidity was significantly lower in the LLR (11.5% vs. 23.7%, *p* = 0.0025) vs. OLR, as was the median LOS (4 vs. 6 days, *p* < 0.001), but there was no difference in OS, DFS, severe postoperative morbidity, or postoperative mortality between the groups [197].

The OSLO-COMET trial randomized 280 patients, 147 LLR and 133 OLR, with a primary endpoint of 30-day postoperative morbidity and secondary endpoints of 5-year OS and DFS. Like the LapOpHuva trial, the patients in the OSLO-COMET trial were well matched with a similar number of lesions, metastasis location, and Iwate complexity scores [199] (a scoring system used to define the complexity of laparoscopic liver resections). The authors found a statistically significant decrease in 30-day postoperative morbidity in the LLR group vs. OLR (19% vs. 31%, *p* = 0.021), lower LOS (53 vs. 96 h, *p* < 0.001), and decreased narcotic requirement (52 vs. 170 mEq, *p* < 0.001), without any differences in perioperative mortality, OR time, EBL, cost-effectiveness, or margin status [198]. A separate quality of life (QOL) analysis was performed, demonstrating that patients who underwent LLR had better social functioning, reduced bodily pain, and increased physical functioning compared to OLR at 1 month postoperatively; improved physical function persisted at the 4-month follow-up [198,200]. The long-term follow-up of this trial demonstrated similar median and 5-year OS between the two groups in an intention-to-treat analysis; independent predictors of worse OS included worse Eastern Cooperative Oncology Group (ECOG) status, positive nodal disease in the colorectal primary, a rectal primary tumor, size of the largest metastasis, and the presence of extrahepatic disease at the time of hepatectomy [201].

These trials highlighted the efficacy of laparoscopy for CRLM; namely, equivalent oncologic outcomes and faster return to work, with reduced periprocedural morbidity, LOS, and perioperative pain. While these trials compared OLR to LLR, more recent studies have evaluated the efficacy of robotic liver resections (RLR).

#### 3.6.2. Robotic Liver Resections

In a propensity-matched cohort of patients with similar tumor burden and liver resections, Kingham et al., found that RLR had shorter OR times (163 min vs. 210 min, *p* = 0.017), reduced EBL (100 vs. 300 mL, *p* < 0.001), and less Pringle use (9% vs. 75%, *p* < 0.001), with similar complication rates and periprocedural mortality [202]. In a systematic review and meta-analysis comparing patients undergoing major hepatectomy via RLR vs. LLR, Ziogas et al., found no difference between the two approaches in the following parameters: overall complications, severe complications, overall mortality, conversion rates, EBL, transfusion rates, R1 rates, OR time, and LOS [203]. Conversely, a more recent multicenter retrospective analysis comparing RLR to LLR found that RLR had lower rates of R1 resection (16.9 vs. 28.8%, *p* = 0.025) and that the benefit of RLR over LLR was increased for more difficult operations or for lesions located in posterosuperior segments [204]. Minimally invasive approaches have also been employed in patients undergoing simultaneous resection of synchronous CRLM and primary colorectal tumors [205]; in fact, a recent Italian national consensus made an endorsement for the minimally invasive approach to simultaneous resection when feasible, using an adapted Delphi method [206]. Interestingly, minimally invasive approaches may also afford a faster return to systemic therapy [207]. Tohme et al., compared 66 patients after propensity-score matching undergoing OLR and MILR. The authors found that MILR was associated with a significantly shorter interval to adjuvant chemotherapy than OLR (median 42 vs. 63 days, *p*  <  0.001) and that the surgical approach remained an independent predictor of earlier return to chemotherapy [207].

The benefits of a minimally invasive approach include improved postoperative morbidity, reduced OR time, improved narcotic requirement/decreased pain, and improved QOL compared to the open approach, without any associated compromise in oncologic outcomes; the differences between minimally invasive techniques is less pronounced, at least in the context of the quality of evidence available at this time, which is limited largely to retrospective studies. It should be noted that, unlike the comparison between OLR and LLR, there are no randomized trials looking specifically at RLR. In recognition of the benefits that a minimally invasive approach provides, the Society of American Gastrointestinal and Endoscopic Surgeons (SAGES) and the Americas Hepato-Pancreato-Biliary Association (AHPBA) have recently released joint conditional recommendations endorsing such approaches for CRLM when appropriate [208].

### 3.7. Ablation

While surgical resection remains the mainstay of curative-intent treatment for CRLM, there is an increasing role for ablative therapies, such as microwave ablation (MWA) and radiofrequency ablation (RFA). This technique can be applied in a percutaneous, laparoscopic, or open approach.

#### 3.7.1. Microwave Ablation and Radiofrequency Ablation

MWA employs a coaxial antenna that emits an alternating electromagnetic field. Polarized molecules within a certain radius of the antennae constantly move to realign themselves with the field, thereby generating kinetic energy that is dissipated as heat. This raises the temperature of the tissue within the ablative zone to over 100 °C, well past the threshold of 60 °C necessary for inducing coagulative necrosis. Unlike older technology, like radiofrequency ablation (RFA), which uses electrical currents to heat tissue, the electromagnetic field generated in MWA permeates through desiccated and normal tissue equally to produce a uniform ablative zone, even in the presence of “heat-sinks”. As such, MWA is being quickly adopted as the new standard for thermal ablation.

#### 3.7.2. Safety and Efficacy of Ablative Therapy

Ablative techniques have been a useful adjunct in the treatment of CRLM, with a low risk of periprocedural complications. Livraghi et al., examined complication rates in 736 patients undergoing MWA for liver lesions through open, laparoscopic, and percutaneous approaches. They found a major complication rate of 2.9% and minor complication rate of 7.3%; the most common complications were periprocedural pain, fever, and asymptomatic pleural effusions not requiring drainage [209]. Along with a low risk of periprocedural complications, MWA ablation has also been shown to successfully treat CRLM in up to 97.5% of cases in some series [210]. The technical success of ablation is measured using local recurrence (LR) at the ablation site, which ranges between 5.2% [210] and 10.9% [211], with rates varying depending on the median follow-up time of the reporting study. A recent article noted that a sizable proportion of patients develop LR 1 year after ablative treatment, thus underpinning the possible limitation of prior studies with shorter follow-up times in accurately assessing ablation efficacy [211]. Nonetheless, most recurrences that occur are at intrahepatic sites other than the ablation zone [210,211]. One phase II trial compared 119 patients undergoing either locoregional treatment for unresectable CRLM using a combination of RFA with or without hepatectomy or systemic chemotherapy alone [212]. The long-term results of this trial demonstrated an 8-year OS of 35.9% in the combined modality arm compared to 8.9% survival seen in the systemic therapy-only arm. All patients (57) in the combined modality arm received RFA, but 47% also underwent concomitant hepatectomy. Given the significant possibility of confounding, attributing the improved survival to ablation only should be done cautiously. Still, the remarkable difference in OS highlights that aggressive local therapy results in improved OS for unresectable CRLM, and that there may be a role for ablative therapies in pursuing this [213].

#### 3.7.3. Limitations

There are some limitations to the use of ablative techniques depending on the size and location of the target lesion. The previous literature established that treating a lesion of >3 cm with ablation was associated with decreased OS and RFS [210], as well as incomplete tumor ablation rates approaching 67% [214]. Karagkounis et al., demonstrated that factors associated with local recurrence on a multivariate analysis included increasing size as a continuous variable (HR 1.04, 95% CI 1.01–1.08; *p* = 0.006) and subcapsular location (HR 2, 95% CI 1.09–3.65; *p* = 0.02); they noted that the cumulative rate of local recurrence at 2 years was 6.8% for tumors ≤10 mm, 12.4% for tumors of 11–20 mm, and 30.2% for tumors >20 mm [211].

Since local ablative therapies are associated with minimal periprocedural complications and have been demonstrated to be effective at treating small tumors, interest has grown in comparing ablative therapy to hepatectomy for resectable CRLM. As such, a phase 3 randomized clinical trial called the COLLISION trial is underway and aims to demonstrate the non-inferiority of ablative therapy (RFA or MWA) to hepatectomy for resectable disease. For now, however, it remains an important adjunct tool for patients undergoing PSS approaches.

### 3.8. Liver Transplantation

#### 3.8.1. History of Liver Transplant for Cancer

Liver transplant (LT) for secondary liver malignancy was largely abandoned in the 1980s, after the early results demonstrated a dismal 5-year OS of 18% [215]; however, as the perioperative care of transplant patients improved dramatically, the expected survival rates of patients undergoing LT for CRLM were also expected to improve since the majority of mortality in those early trials was for reasons other than tumor progression.

#### 3.8.2. SECA-I

The SECA-1 trial was a nonrandomized, prospective pilot study of 21 patients who underwent LT for unresectable CRLM [216]. Exclusion criteria were extrahepatic disease, weight loss > 10%, and other standard contraindications for LT. Median age was 56 years old, 57% of patients were on second-line therapy or further, and the median CRS was 3. Through a median follow-up of 27 months, the estimated 5-year OS was 60%, and the median DFS was 8 months; the most common site of recurrence was the lung (80%), followed by liver recurrence. The authors proposed a criterion for selecting patients for LT based on the predictive factors of survival from this trial, called the Oslo criteria; its components are depicted in Table 4 below. Of note, four patients (19%) had vascular complications from LT, of which three needed re-transplantation. Additionally, five patients (24%) required reoperation for hemorrhage.

A recent long-term update to the original SECA-1 trial included encouraging 10-year follow-up data [217]. The actual 10-year OS was 26.1%, while median DFS was 10 months. All patients recurred, and the most common site of recurrence was the lung; of the patients who survived for 10 years, all underwent lung resections. Survival varied greatly amongst the different patient subgroups. The Oslo score of 3–4 patients had a median OS of 26 months, and all had died by 86 months; meanwhile, Oslo scores 0–1 had a 10-year OS of 50%, and Oslo score 2 had a 10-year survival of 33%.

The results of the SECA-I trial were used as justification for the SECA-II trial, which had much stricter inclusion criteria to prolong survival further. The main inclusion criteria for both trials are listed in Table 5 below.

#### 3.8.3. SECA-II

SECA-II was a prospective trial that examined 15 patients with stricter inclusion criteria [218]. The baseline characteristics included a median age of 59, 13 left-sided primary tumors, median CRS 2, median Oslo score 1, and time from primary resection to LT of 22 months. At the time of the CRLM diagnosis, the median tumor number and size was 12 and 45 mm, respectively, with a decrease to 5 and 26 mm, respectively, prior to LT; similarly, the median Fong CRS at diagnosis was 3, and at LT, it was 2. The median times from diagnosis to LT and primary resection to LT were 24 and 22.6 months, respectively. With a median follow-up of 36 months, the 5-year OS was reported as 83%, and the 5-year DFS was 35%. Eight patients developed recurrence, of which two developed oligometastatic disease and received palliative chemotherapy. Six patients developed pulmonary metastasis with six lung resections performed in five patients. Severe complications (Clavien–Dindo grade ≥ IIIa) occurred in 47% of patients. DFS was expectedly shorter in patients with higher tumor burden (>8 lesions or Fong CRS 3–4 vs. 1–2).

#### 3.8.4. Interpretation of the Data

The results of the SECA-I and II trials have had an enormous impact on the field of transplant for CRLM, with several ongoing clinical trials now accruing patients. However, the results of the data should be viewed cautiously. For the SECA-II trial, the definition of unresectable disease was not explicitly stated, and there were several patients who had <5 lesions, as well as all patients having a remarkable response to pre-LT therapy. Additionally, the median follow-up time was limited to 36 months, and only six patients had 3-year follow-up data, while one had 5-year data at the time of publication. The authors also make comparisons between their cohort and larger chemotherapy trials, where the burden of disease is significantly different (given that those patients often have extrahepatic disease as well) and the inclusion criteria are far less strict; as such, comparisons between trials are imperfect and should be met with hesitation. The patients in the SECA-II cohort are so heavily selected, such that they would likely have good outcomes regardless of intervention. Finally, the true benefit of LT in CRLM can only be determined by comparing a similarly selected population of patients who undergo LT versus the current standard of care.

While there is almost certainly a role for LT in the treatment of CRLM, it needs to be more clearly defined; currently, LT for CRLM should only be performed in the setting of a strict study design to generate more prospective data.

### 3.9. Literature Research Strategy

As a narrative review, the authors independently reviewed the subject matter and agreed upon an outline that covered the desired material. Once decided upon, the authors looked for the highest-quality and most relevant evidence available pertaining to the message the authors wished to convey with each topic. In this regard, randomized trials served as the gold-standard reference. In the absence of prospective studies, systematic reviews and meta-analysis were reviewed and then served as a means for identifying other highly cited and impactful articles. This manuscript is not meant to serve as a systematic review but rather represents the views of the authors as they pertain to the contemporary surgical management of CRLM.

## 4. Conclusions

The contemporary management of CRLM involves a complex interplay between a variety of perioperative and operative considerations, all of which are constantly evolving. The roles of chemotherapeutics and dosing strategies, as well as specific surgical procedures, remain to be clarified using prospective data from future trials. However, currently, surgical resection remains the gold-standard treatment for CRLM as a potentially curative therapy.

## Figures and Tables

**Figure 1 cancers-16-00941-f001:**
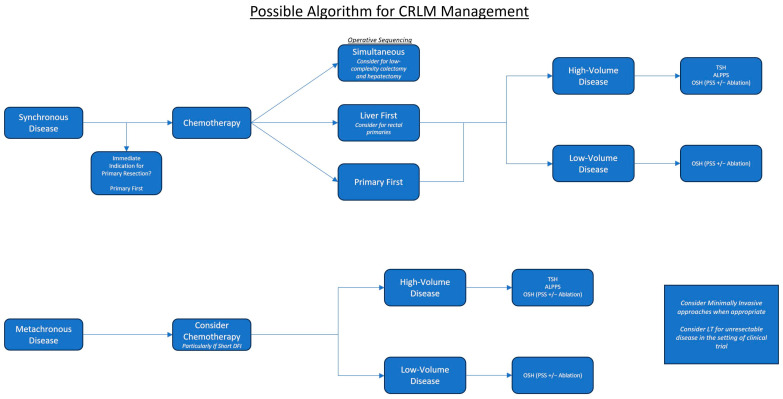
Flow diagram of surgical management of CRLM. Abbreviations: CRLM—colorectal liver metastasis; TSH—two-stage hepatectomy; ALPPS—associating liver partition and portal vein ligation for staged hepatectomy; OSH—one-stage hepatectomy; PSS—parenchymal-sparing surgery, DFI—disease-free interval; LT—liver transplantation.

**Table 2 cancers-16-00941-t002:** RAS Mutational Risk Score [52] vs. Fong Clinical Risk Score [34].

Fong Clinical Risk Score (0–5)	RAS Mutational Risk Score (0–3)
Largest tumor > 5 cm Disease-free interval between primary and diagnosis of CRLM of <12 months Number of metastases > 1 Preoperative CEA > 200 ug/L Node-positive primary	Largest tumor > 5 cm KRAS Mutational Status Node-positive primary

Abbreviations: CEA—carcinoembryonic antigen; CRLM—colorectal liver metastases; KRAS—Kirsten rat sarcoma viral oncogene homolog.

**Table 3 cancers-16-00941-t003:** Modified table from Xu et al., depicting influence of R0 and R1 resections on 5-year OS stratified according to RAS status and response to chemotherapy [163].

Tumor Biology	RAS Status	Response to Chemotherapy (>50%)	5-Year OS		*p*-Value
			R0	R1 (<1 mm)	
Good	*wt*	Yes	66.4%	65.2%	0.884
Bad	*mt*	Yes	58.5%	48.7%	**0.043**
	*wt*	No	24.6%	11.1%	**0.024**
	*mt*	No	19.5%	0%	**0.022**

Abbreviations: RAS—Kirsten rat sarcoma viral oncogene homolog; OS—overall survival; *mt*—mutant; *wt*—wildtype.

**Table 4 cancers-16-00941-t004:** Oslo Criteria [216] vs. Fong Clinical Risk Score [34].

Oslo Score (0–4)	Fong Clinical Risk Score (0–5)
Largest tumor > 5.5 cm Less than 2-year interval between primary resection and LT Progressive disease at the time of LT Preoperative CEA > 80 ug/L	Largest tumor > 5 cm Disease-free interval between primary and diagnosis of CRLM of <12 months Number of metastases > 1 Preoperative CEA > 200 ug/L Node-positive primary

Abbreviations: LT—liver transplant; CEA—carcinoembryonic antigen; CRLM—colorectal liver metastases.

**Table 5 cancers-16-00941-t005:** Inclusion criteria from SECA-I [216] and SECA-II [218] trials.

SECA-I	SECA-II
- Histologically confirmed colorectal adenocarcinoma - No extrahepatic disease or local recurrence via PET/CT - No extrahepatic disease or local recurrence via CT or MRI - Completed radical excision of primary tumor - ECOG score 0 or 1 - Minimum of 6 weeks of chemotherapy regardless of response	- Histologically confirmed colorectal adenocarcinoma - No extrahepatic disease or local recurrence via PET/CT - No extrahepatic disease or local recurrence via CT or MRI chest/abdomen/pelvis within 4 weeks of transplant meeting - No local recurrence via CT colonography or colonoscopy within 1 year of transplant meeting - ECOG score 0 or 1 - Labs: Hb > 10, neutrophils > 1 (after any G-CSF), Bilirubin < 2× upper limit of normal, AST/ALT < 5× upper limit of normal, Creatinine < 1.25× upper limit of normal, Albumin above lower limit of normal - Negative margins of primary and at least ≥ 2 mm circumferential resection margin of rectal primaries - Unresectable liver metastasis, either upfront or after prior hepatectomy - Received first-line chemotherapy and need at least 10% RECIST response to any chemotherapy - Before chemotherapy, no lesion > 10 cm; and if >30 lesions, no lesion > 5 cm and need at least 30% response by RECIST - If <10% RECIST response, can be included after TACE or Y90 if >20% RECIST response - At least 1 year between CRC diagnosis and listing date for LT

Abbreviations: PET—positron emission tomographic; CT—computed tomographic; MRI—magnetic resonance imaging; ECOG—Eastern Cooperative Oncology Group; Hb—hemoglobin; G-CSF—granulocyte colony stimulating factor; AST—aspartate aminotransferase; ALT—alanine transaminase; RECIST—Response Evaluation Criteria in Solid Tumors; TACE—transarterial chemoembolization; Y90—Yttrium-90 radioembolization; LT—liver transplant.

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
