# Peer review of "Contemporary Surgical Management of Colorectal Liver Metastases"

_cancers, 2024, doi:10.3390/cancers16050941_

Round 1
Reviewer 1 Report
Comments and Suggestions for Authors
This is a conprehensive review for the current status of CRC liver mets surgical management.
Detailed review and discussion were done in every item. However, the authors can put one summary figure or table for the current overview of the surgical managements for the disease.
Reviewer 2 Report
Comments and Suggestions for Authors
This is a very fine review on surgical treatment options in CRLM. Nearly all aspects are addressed.
The only item, that should be added to the chapter "duration of chemotherapy", is the topic of liver metastases that are not longer detected in imaging after some cycles of chemotherapy.
Reviewer 3 Report
Comments and Suggestions for Authors
I have read with interest this comprehensive review on the surgical management of colorectal liver metastases. I have found this paper well written, well organized and readable and I think it will be of interest to the readership of Cancers. No changes are suggested.
Reviewer 4 Report
Comments and Suggestions for Authors
dear Author thank you for submitting your paper to cancers. The paper is well written and treats an actual and interesting topic, however I have few minor comments.
The paper summarizes the whole treatment strategies for CRLM, but there are two weak points inn my point of view.
First of all, despite it is not a systematic review, but only a narrative review, it would be nice if you add a dedicated paragraph to literature research strategy helping the readers to understand the strength and methodology of your literature research.
Secondly, concerning minimally invasive approach in my opinion you missed important references like: Masetti M, Fallani G, Ratti F, Ferrero A, Giuliante F, Cillo U, Guglielmi A, Ettorre GM, Torzilli G, Vincenti L, Ercolani G, Cipressi C, Lombardi R, Aldrighetti L, Jovine E. Minimally invasive treatment of colorectal liver metastases: does robotic surgery provide any technical advantages over laparoscopy? A multicenter analysis from the IGoMILS (Italian Group of Minimally Invasive Liver Surgery) registry. Updates Surg. 2022 Apr;74(2):535-545. doi: 10.1007/s13304-022-01245-1. Epub 2022 Jan 31. PMID: 35099776.
Ruzzenente A, Ciangherotti A, Aldrighetti L, Ettorre GM, De Carlis L, Ferrero A, Dalla Valle R, Tisone G, Guglielmi A; IgoMILS – Sg1 Group. Technical feasibility and short-term outcomes of laparoscopic isolated caudate lobe resection: an IgoMILS (Italian Group of Minimally Invasive Liver Surgery) registry-based study. Surg Endosc. 2022 Feb;36(2):1490-1499. doi: 10.1007/s00464-021-08434-w. Epub 2021 Mar 31. PMID: 33788031; PMCID: PMC8758628.
Rocca A, Cipriani F, Belli G, Berti S, Boggi U, Bottino V, Cillo U, Cescon M, Cimino M, Corcione F, De Carlis L, Degiuli M, De Paolis P, De Rose AM, D'Ugo D, Di Benedetto F, Elmore U, Ercolani G, Ettorre GM, Ferrero A, Filauro M, Giuliante F, Gruttadauria S, Guglielmi A, Izzo F, Jovine E, Laurenzi A, Marchegiani F, Marini P, Massani M, Mazzaferro V, Mineccia M, Minni F, Muratore A, Nicosia S, Pellicci R, Rosati R, Russolillo N, Spinelli A, Spolverato G, Torzilli G, Vennarecci G, Viganò L, Vincenti L, Delrio P, Calise F, Aldrighetti L. The Italian Consensus on minimally invasive simultaneous resections for synchronous liver metastasis and primary colorectal cancer: A Delphi methodology. Updates Surg. 2021 Aug;73(4):1247-1265. doi: 10.1007/s13304-021-01100-9. Epub 2021 Jun 5. PMID: 34089501.
Endo Y, Alaimo L, Moazzam Z, Woldesenbet S, Lima HA, Munir MM, Shaikh CF, Yang J, Azap L, Katayama E, Guglielmi A, Ruzzenente A, Aldrighetti L, Alexandrescu S, Kitago M, Poultsides G, Sasaki K, Aucejo F, Pawlik TM. Postoperative morbidity after simultaneous versus staged resection of synchronous colorectal liver metastases: Impact of hepatic tumor burden. Surgery. 2023 Nov 23:S0039-6060(23)00811-5. doi: 10.1016/j.surg.2023.10.019. Epub ahead of print. PMID: 38001013.
Please add contents from these references to the paper.
Reviewer 5 Report
Comments and Suggestions for Authors
This narrative review aimed to discuss the aspects of the contemporary peri- and post-operative management of patients with colorectal liver metastases (CRLM). The manuscript is well written and very comprehensive; it addressed all the most important elements which must be considered when dealing with CRLM patients.
Some issues should be kindly addressed:
1) In table 1 among the “anatomic factors of resectability”, the ability to achieve a R0 resection could be added to easier the reader’s comprehension (exceptions to this golden rule are well described in other sections of the manuscript).
2) At line 141, you correctly state that the old risk scores model considered clinicopathologic factors but not biological ones. It would be useful to underline another limitation of such risk scores (like the Fong one), which is that those scores were developed in an era when effective chemotherapy still not existed (FOLFOX introduced in clinical practice in the early 2000’s), and that almost none of the patients included in those series received preoperative/neoadjuvant treatments. Nowadays many CRLM patients receive preoperative systemic treatment (based on oncologic more than anatomic risk factors), questioning the applicability of the old risk scores. Lastly, non-response or progression during preoperative chemotherapy could be considered a surrogate marker of biological aggressiveness of the disease.
3) In the section “operative sequencing for synchronous disease” (line 157), you should precise that irrespective of the decision to which site resect first, most of the patients should receive preoperative treatment (i.e., neoadjuvant or conversion intent). You should also precise in which conditions an upfront resective approach (and not following a systemic treatment) could be considered (symptomatic primary, obstruction, bleeding, etc.).
4) Specifically regarding the “liver-first” approach, would you please cite the first landmark paper describing this strategy (Mentha G, Majno PE, Andres A, Rubbia-Brandt L, Morel P, Roth AD. Neoadjuvant chemotherapy and resection of advanced synchronous liver metastases before treatment of the colorectal primary. Br J Surg. 2006 Jul;93(7):872-8. doi: 10.1002/bjs.5346. PMID: 16671066.), and the subsequent work validating it on a large scale (Andres A, Toso C, Adam R, Barroso E, Hubert C, Capussotti L, Gerstel E, Roth A, Majno PE, Mentha G. A survival analysis of the liver-first reversed management of advanced simultaneous colorectal liver metastases: a LiverMetSurvey-based study. Ann Surg. 2012 Nov;256(5):772-8; discussion 778-9. doi: 10.1097/SLA.0b013e3182734423. PMID: 23095621.)
5) In the “chemotherapy sequencing” paragraph (line 227), first you state that preoperative treatment allows to test tumor biology eventually avoiding non-beneficial surgery in case of progression. Shortly after (line 235) you state that in borderline resectable CRLM neoadjuvant chemotherapy may lead to progression into unresectability, thus in this specific setting maybe adjuvant treatment only would be a better option, potentially creating confusion for the reader. This introductive paragraph should be clarified: there is preoperative/neoadjuvant/induction treatment (in case of upfront resectable or borderline resectable disease), and conversion treatment (initially unresectable CRLM). While the latter scenario leaves no room for discussion, in the setting of upfront resectable and borderline resectable disease the use of preoperative chemotherapy is much more debated (and you correctly describe it). Then there is post-operative/adjuvant treatment (see below).
6) In the paragraph “adjuvant therapy” (line 240) you do not mention the JCOG0603 trial from Japan (Kanemitsu Y, Shimizu Y, Mizusawa J, Inaba Y, Hamaguchi T, Shida D, Ohue M, Komori K, Shiomi A, Shiozawa M, Watanabe J, Suto T, Kinugasa Y, Takii Y, Bando H, Kobatake T, Inomata M, Shimada Y, Katayama H, Fukuda H; JCOG Colorectal Cancer Study Group. Hepatectomy Followed by mFOLFOX6 Versus Hepatectomy Alone for Liver-Only Metastatic Colorectal Cancer (JCOG0603): A Phase II or III Randomized Controlled Trial. J Clin Oncol. 2021 Dec 1;39(34):3789-3799. doi: 10.1200/JCO.21.01032. Epub 2021 Sep 14. PMID: 34520230.). This trial (to my knowledge) is the most recent available addressing the use of post-operative systemic treatment in CRLM patients and confirmed an advantage in disease-free survival (but not in overall survival) for hepatectomy followed by mFOLFOX6 compared to hepatectomy alone. This trial was one of the bases for ASCO 2023 guidelines on metastatic colorectal cancer (see Table 11 of ASCO 2023 guidelines) and I believe it should be mentioned. That said, lots of retrospective studies reported better OS with the use of adjuvant treatment compared to hepatectomy alone, maybe you could include in the discussion some recent meta-analysis.
7) “Perioperative therapy” paragraph: according to what above mentioned, the initial statement (line 260) could be too strong (there is level 1b evidence in DFS, but not in OS).
8) Line 335: you state that in EPOC trial (EORTC 40983) by Nordlinger, et al. did not include data regarding chemotherapy-related toxicity. This seems to me incorrect, such data are provided in Table 2 and 3 of the trial. Please correct.
9) Line 419: the correct name is “AZOULAY”, please correct.
10) “Resection margin” paragraph (line 469), I would suggest moving the part related to R1 vascular (Viganò, Torzilli) at the end of this paragraph instead than in the “parenchymal-sparing surgery” paragraph. I agree that R1vasc is one of the elements of PSS but it is worthy of mention when you discuss about the length of the margins, again possibly mentioning the quality of the evidence available on R1vasc. Similarly, I wonder if E-OSH (line 648) would be better discussed following TSH and ALPPS rather than in PSS section, however, this last one is just a suggestion.
11) “Robotic liver resection” paragraph (line 697). I suggest to clarify which is the level of evidence currently available, particularly because – in contrast to laparoscopic approach – there is no randomized trials available.
Thanks for addressing the abovementioned points, I wish to thank once again the authors for their effort in elaborating this comprehensive manuscript.
Comments on the Quality of English Language
This narrative review aimed to discuss the aspects of the contemporary peri- and post-operative management of patients with colorectal liver metastases (CRLM). The manuscript is well written and very comprehensive; it addressed all the most important elements which must be considered when dealing with CRLM patients.
Some issues should be kindly addressed:
1) In table 1 among the “anatomic factors of resectability”, the ability to achieve a R0 resection could be added to easier the reader’s comprehension (exceptions to this golden rule are well described in other sections of the manuscript).
2) At line 141, you correctly state that the old risk scores model considered clinicopathologic factors but not biological ones. It would be useful to underline another limitation of such risk scores (like the Fong one), which is that those scores were developed in an era when effective chemotherapy still not existed (FOLFOX introduced in clinical practice in the early 2000’s), and that almost none of the patients included in those series received preoperative/neoadjuvant treatments. Nowadays many CRLM patients receive preoperative systemic treatment (based on oncologic more than anatomic risk factors), questioning the applicability of the old risk scores. Lastly, non-response or progression during preoperative chemotherapy could be considered a surrogate marker of biological aggressiveness of the disease.
3) In the section “operative sequencing for synchronous disease” (line 157), you should precise that irrespective of the decision to which site resect first, most of the patients should receive preoperative treatment (i.e., neoadjuvant or conversion intent). You should also precise in which conditions an upfront resective approach (and not following a systemic treatment) could be considered (symptomatic primary, obstruction, bleeding, etc.).
4) Specifically regarding the “liver-first” approach, would you please cite the first landmark paper describing this strategy (Mentha G, Majno PE, Andres A, Rubbia-Brandt L, Morel P, Roth AD. Neoadjuvant chemotherapy and resection of advanced synchronous liver metastases before treatment of the colorectal primary. Br J Surg. 2006 Jul;93(7):872-8. doi: 10.1002/bjs.5346. PMID: 16671066.), and the subsequent work validating it on a large scale (Andres A, Toso C, Adam R, Barroso E, Hubert C, Capussotti L, Gerstel E, Roth A, Majno PE, Mentha G. A survival analysis of the liver-first reversed management of advanced simultaneous colorectal liver metastases: a LiverMetSurvey-based study. Ann Surg. 2012 Nov;256(5):772-8; discussion 778-9. doi: 10.1097/SLA.0b013e3182734423. PMID: 23095621.)
5) In the “chemotherapy sequencing” paragraph (line 227), first you state that preoperative treatment allows to test tumor biology eventually avoiding non-beneficial surgery in case of progression. Shortly after (line 235) you state that in borderline resectable CRLM neoadjuvant chemotherapy may lead to progression into unresectability, thus in this specific setting maybe adjuvant treatment only would be a better option, potentially creating confusion for the reader. This introductive paragraph should be clarified: there is preoperative/neoadjuvant/induction treatment (in case of upfront resectable or borderline resectable disease), and conversion treatment (initially unresectable CRLM). While the latter scenario leaves no room for discussion, in the setting of upfront resectable and borderline resectable disease the use of preoperative chemotherapy is much more debated (and you correctly describe it). Then there is post-operative/adjuvant treatment (see below).
6) In the paragraph “adjuvant therapy” (line 240) you do not mention the JCOG0603 trial from Japan (Kanemitsu Y, Shimizu Y, Mizusawa J, Inaba Y, Hamaguchi T, Shida D, Ohue M, Komori K, Shiomi A, Shiozawa M, Watanabe J, Suto T, Kinugasa Y, Takii Y, Bando H, Kobatake T, Inomata M, Shimada Y, Katayama H, Fukuda H; JCOG Colorectal Cancer Study Group. Hepatectomy Followed by mFOLFOX6 Versus Hepatectomy Alone for Liver-Only Metastatic Colorectal Cancer (JCOG0603): A Phase II or III Randomized Controlled Trial. J Clin Oncol. 2021 Dec 1;39(34):3789-3799. doi: 10.1200/JCO.21.01032. Epub 2021 Sep 14. PMID: 34520230.). This trial (to my knowledge) is the most recent available addressing the use of post-operative systemic treatment in CRLM patients and confirmed an advantage in disease-free survival (but not in overall survival) for hepatectomy followed by mFOLFOX6 compared to hepatectomy alone. This trial was one of the bases for ASCO 2023 guidelines on metastatic colorectal cancer (see Table 11 of ASCO 2023 guidelines) and I believe it should be mentioned. That said, lots of retrospective studies reported better OS with the use of adjuvant treatment compared to hepatectomy alone, maybe you could include in the discussion some recent meta-analysis.
7) “Perioperative therapy” paragraph: according to what above mentioned, the initial statement (line 260) could be too strong (there is level 1b evidence in DFS, but not in OS).
8) Line 335: you state that in EPOC trial (EORTC 40983) by Nordlinger, et al. did not include data regarding chemotherapy-related toxicity. This seems to me incorrect, such data are provided in Table 2 and 3 of the trial. Please correct.
9) Line 419: the correct name is “AZOULAY”, please correct.
10) “Resection margin” paragraph (line 469), I would suggest moving the part related to R1 vascular (Viganò, Torzilli) at the end of this paragraph instead than in the “parenchymal-sparing surgery” paragraph. I agree that R1vasc is one of the elements of PSS but it is worthy of mention when you discuss about the length of the margins, again possibly mentioning the quality of the evidence available on R1vasc. Similarly, I wonder if E-OSH (line 648) would be better discussed following TSH and ALPPS rather than in PSS section, however, this last one is just a suggestion.
11) “Robotic liver resection” paragraph (line 697). I suggest to clarify which is the level of evidence currently available, particularly because – in contrast to laparoscopic approach – there is no randomized trials available.
Thanks for addressing the abovementioned points, I wish to thank once again the authors for their effort in elaborating this comprehensive manuscript.
Some issues should be kindly addressed:
1) In table 1 among the “anatomic factors of resectability”, the ability to achieve a R0 resection could be added to easier the reader’s comprehension (exceptions to this golden rule are well described in other sections of the manuscript).
2) At line 141, you correctly state that the old risk scores model considered clinicopathologic factors but not biological ones. It would be useful to underline another limitation of such risk scores (like the Fong one), which is that those scores were developed in an era when effective chemotherapy still not existed (FOLFOX introduced in clinical practice in the early 2000’s), and that almost none of the patients included in those series received preoperative/neoadjuvant treatments. Nowadays many CRLM patients receive preoperative systemic treatment (based on oncologic more than anatomic risk factors), questioning the applicability of the old risk scores. Lastly, non-response or progression during preoperative chemotherapy could be considered a surrogate marker of biological aggressiveness of the disease.
3) In the section “operative sequencing for synchronous disease” (line 157), you should precise that irrespective of the decision to which site resect first, most of the patients should receive preoperative treatment (i.e., neoadjuvant or conversion intent). You should also precise in which conditions an upfront resective approach (and not following a systemic treatment) could be considered (symptomatic primary, obstruction, bleeding, etc.).
4) Specifically regarding the “liver-first” approach, would you please cite the first landmark paper describing this strategy (Mentha G, Majno PE, Andres A, Rubbia-Brandt L, Morel P, Roth AD. Neoadjuvant chemotherapy and resection of advanced synchronous liver metastases before treatment of the colorectal primary. Br J Surg. 2006 Jul;93(7):872-8. doi: 10.1002/bjs.5346. PMID: 16671066.), and the subsequent work validating it on a large scale (Andres A, Toso C, Adam R, Barroso E, Hubert C, Capussotti L, Gerstel E, Roth A, Majno PE, Mentha G. A survival analysis of the liver-first reversed management of advanced simultaneous colorectal liver metastases: a LiverMetSurvey-based study. Ann Surg. 2012 Nov;256(5):772-8; discussion 778-9. doi: 10.1097/SLA.0b013e3182734423. PMID: 23095621.)
5) In the “chemotherapy sequencing” paragraph (line 227), first you state that preoperative treatment allows to test tumor biology eventually avoiding non-beneficial surgery in case of progression. Shortly after (line 235) you state that in borderline resectable CRLM neoadjuvant chemotherapy may lead to progression into unresectability, thus in this specific setting maybe adjuvant treatment only would be a better option, potentially creating confusion for the reader. This introductive paragraph should be clarified: there is preoperative/neoadjuvant/induction treatment (in case of upfront resectable or borderline resectable disease), and conversion treatment (initially unresectable CRLM). While the latter scenario leaves no room for discussion, in the setting of upfront resectable and borderline resectable disease the use of preoperative chemotherapy is much more debated (and you correctly describe it). Then there is post-operative/adjuvant treatment (see below).
6) In the paragraph “adjuvant therapy” (line 240) you do not mention the JCOG0603 trial from Japan (Kanemitsu Y, Shimizu Y, Mizusawa J, Inaba Y, Hamaguchi T, Shida D, Ohue M, Komori K, Shiomi A, Shiozawa M, Watanabe J, Suto T, Kinugasa Y, Takii Y, Bando H, Kobatake T, Inomata M, Shimada Y, Katayama H, Fukuda H; JCOG Colorectal Cancer Study Group. Hepatectomy Followed by mFOLFOX6 Versus Hepatectomy Alone for Liver-Only Metastatic Colorectal Cancer (JCOG0603): A Phase II or III Randomized Controlled Trial. J Clin Oncol. 2021 Dec 1;39(34):3789-3799. doi: 10.1200/JCO.21.01032. Epub 2021 Sep 14. PMID: 34520230.). This trial (to my knowledge) is the most recent available addressing the use of post-operative systemic treatment in CRLM patients and confirmed an advantage in disease-free survival (but not in overall survival) for hepatectomy followed by mFOLFOX6 compared to hepatectomy alone. This trial was one of the bases for ASCO 2023 guidelines on metastatic colorectal cancer (see Table 11 of ASCO 2023 guidelines) and I believe it should be mentioned. That said, lots of retrospective studies reported better OS with the use of adjuvant treatment compared to hepatectomy alone, maybe you could include in the discussion some recent meta-analysis.
7) “Perioperative therapy” paragraph: according to what above mentioned, the initial statement (line 260) could be too strong (there is level 1b evidence in DFS, but not in OS).
8) Line 335: you state that in EPOC trial (EORTC 40983) by Nordlinger, et al. did not include data regarding chemotherapy-related toxicity. This seems to me incorrect, such data are provided in Table 2 and 3 of the trial. Please correct.
9) Line 419: the correct name is “AZOULAY”, please correct.
10) “Resection margin” paragraph (line 469), I would suggest moving the part related to R1 vascular (Viganò, Torzilli) at the end of this paragraph instead than in the “parenchymal-sparing surgery” paragraph. I agree that R1vasc is one of the elements of PSS but it is worthy of mention when you discuss about the length of the margins, again possibly mentioning the quality of the evidence available on R1vasc. Similarly, I wonder if E-OSH (line 648) would be better discussed following TSH and ALPPS rather than in PSS section, however, this last one is just a suggestion.
11) “Robotic liver resection” paragraph (line 697). I suggest to clarify which is the level of evidence currently available, particularly because – in contrast to laparoscopic approach – there is no randomized trials available.
Thanks for addressing the abovementioned points, I wish to thank once again the authors for their effort in elaborating this comprehensive manuscript.
This narrative review aimed to discuss the aspects of the contemporary peri- and post-operative management of patients with colorectal liver metastases (CRLM). The manuscript is well written and very comprehensive; it addressed all the most important elements which must be considered when dealing with CRLM patients.
Some issues should be kindly addressed:
1) In table 1 among the “anatomic factors of resectability”, the ability to achieve a R0 resection could be added to easier the reader’s comprehension (exceptions to this golden rule are well described in other sections of the manuscript).
2) At line 141, you correctly state that the old risk scores model considered clinicopathologic factors but not biological ones. It would be useful to underline another limitation of such risk scores (like the Fong one), which is that those scores were developed in an era when effective chemotherapy still not existed (FOLFOX introduced in clinical practice in the early 2000’s), and that almost none of the patients included in those series received preoperative/neoadjuvant treatments. Nowadays many CRLM patients receive preoperative systemic treatment (based on oncologic more than anatomic risk factors), questioning the applicability of the old risk scores. Lastly, non-response or progression during preoperative chemotherapy could be considered a surrogate marker of biological aggressiveness of the disease.
3) In the section “operative sequencing for synchronous disease” (line 157), you should precise that irrespective of the decision to which site resect first, most of the patients should receive preoperative treatment (i.e., neoadjuvant or conversion intent). You should also precise in which conditions an upfront resective approach (and not following a systemic treatment) could be considered (symptomatic primary, obstruction, bleeding, etc.).
4) Specifically regarding the “liver-first” approach, would you please cite the first landmark paper describing this strategy (Mentha G, Majno PE, Andres A, Rubbia-Brandt L, Morel P, Roth AD. Neoadjuvant chemotherapy and resection of advanced synchronous liver metastases before treatment of the colorectal primary. Br J Surg. 2006 Jul;93(7):872-8. doi: 10.1002/bjs.5346. PMID: 16671066.), and the subsequent work validating it on a large scale (Andres A, Toso C, Adam R, Barroso E, Hubert C, Capussotti L, Gerstel E, Roth A, Majno PE, Mentha G. A survival analysis of the liver-first reversed management of advanced simultaneous colorectal liver metastases: a LiverMetSurvey-based study. Ann Surg. 2012 Nov;256(5):772-8; discussion 778-9. doi: 10.1097/SLA.0b013e3182734423. PMID: 23095621.)
5) In the “chemotherapy sequencing” paragraph (line 227), first you state that preoperative treatment allows to test tumor biology eventually avoiding non-beneficial surgery in case of progression. Shortly after (line 235) you state that in borderline resectable CRLM neoadjuvant chemotherapy may lead to progression into unresectability, thus in this specific setting maybe adjuvant treatment only would be a better option, potentially creating confusion for the reader. This introductive paragraph should be clarified: there is preoperative/neoadjuvant/induction treatment (in case of upfront resectable or borderline resectable disease), and conversion treatment (initially unresectable CRLM). While the latter scenario leaves no room for discussion, in the setting of upfront resectable and borderline resectable disease the use of preoperative chemotherapy is much more debated (and you correctly describe it). Then there is post-operative/adjuvant treatment (see below).
6) In the paragraph “adjuvant therapy” (line 240) you do not mention the JCOG0603 trial from Japan (Kanemitsu Y, Shimizu Y, Mizusawa J, Inaba Y, Hamaguchi T, Shida D, Ohue M, Komori K, Shiomi A, Shiozawa M, Watanabe J, Suto T, Kinugasa Y, Takii Y, Bando H, Kobatake T, Inomata M, Shimada Y, Katayama H, Fukuda H; JCOG Colorectal Cancer Study Group. Hepatectomy Followed by mFOLFOX6 Versus Hepatectomy Alone for Liver-Only Metastatic Colorectal Cancer (JCOG0603): A Phase II or III Randomized Controlled Trial. J Clin Oncol. 2021 Dec 1;39(34):3789-3799. doi: 10.1200/JCO.21.01032. Epub 2021 Sep 14. PMID: 34520230.). This trial (to my knowledge) is the most recent available addressing the use of post-operative systemic treatment in CRLM patients and confirmed an advantage in disease-free survival (but not in overall survival) for hepatectomy followed by mFOLFOX6 compared to hepatectomy alone. This trial was one of the bases for ASCO 2023 guidelines on metastatic colorectal cancer (see Table 11 of ASCO 2023 guidelines) and I believe it should be mentioned. That said, lots of retrospective studies reported better OS with the use of adjuvant treatment compared to hepatectomy alone, maybe you could include in the discussion some recent meta-analysis.
7) “Perioperative therapy” paragraph: according to what above mentioned, the initial statement (line 260) could be too strong (there is level 1b evidence in DFS, but not in OS).
8) Line 335: you state that in EPOC trial (EORTC 40983) by Nordlinger, et al. did not include data regarding chemotherapy-related toxicity. This seems to me incorrect, such data are provided in Table 2 and 3 of the trial. Please correct.
9) Line 419: the correct name is “AZOULAY”, please correct.
10) “Resection margin” paragraph (line 469), I would suggest moving the part related to R1 vascular (Viganò, Torzilli) at the end of this paragraph instead than in the “parenchymal-sparing surgery” paragraph. I agree that R1vasc is one of the elements of PSS but it is worthy of mention when you discuss about the length of the margins, again possibly mentioning the quality of the evidence available on R1vasc. Similarly, I wonder if E-OSH (line 648) would be better discussed following TSH and ALPPS rather than in PSS section, however, this last one is just a suggestion.
11) “Robotic liver resection” paragraph (line 697). I suggest to clarify which is the level of evidence currently available, particularly because – in contrast to laparoscopic approach – there is no randomized trials available.
Thanks for addressing the abovementioned points, I wish to thank once again the authors for their effort in elaborating this comprehensive manuscript.
This narrative review aimed to discuss the aspects of the contemporary peri- and post-operative management of patients with colorectal liver metastases (CRLM). The manuscript is well written and very comprehensive; it addressed all the most important elements which must be considered when dealing with CRLM patients.
Some issues should be kindly addressed:
1) In table 1 among the “anatomic factors of resectability”, the ability to achieve a R0 resection could be added to easier the reader’s comprehension (exceptions to this golden rule are well described in other sections of the manuscript).
2) At line 141, you correctly state that the old risk scores model considered clinicopathologic factors but not biological ones. It would be useful to underline another limitation of such risk scores (like the Fong one), which is that those scores were developed in an era when effective chemotherapy still not existed (FOLFOX introduced in clinical practice in the early 2000’s), and that almost none of the patients included in those series received preoperative/neoadjuvant treatments. Nowadays many CRLM patients receive preoperative systemic treatment (based on oncologic more than anatomic risk factors), questioning the applicability of the old risk scores. Lastly, non-response or progression during preoperative chemotherapy could be considered a surrogate marker of biological aggressiveness of the disease.
3) In the section “operative sequencing for synchronous disease” (line 157), you should precise that irrespective of the decision to which site resect first, most of the patients should receive preoperative treatment (i.e., neoadjuvant or conversion intent). You should also precise in which conditions an upfront resective approach (and not following a systemic treatment) could be considered (symptomatic primary, obstruction, bleeding, etc.).
4) Specifically regarding the “liver-first” approach, would you please cite the first landmark paper describing this strategy (Mentha G, Majno PE, Andres A, Rubbia-Brandt L, Morel P, Roth AD. Neoadjuvant chemotherapy and resection of advanced synchronous liver metastases before treatment of the colorectal primary. Br J Surg. 2006 Jul;93(7):872-8. doi: 10.1002/bjs.5346. PMID: 16671066.), and the subsequent work validating it on a large scale (Andres A, Toso C, Adam R, Barroso E, Hubert C, Capussotti L, Gerstel E, Roth A, Majno PE, Mentha G. A survival analysis of the liver-first reversed management of advanced simultaneous colorectal liver metastases: a LiverMetSurvey-based study. Ann Surg. 2012 Nov;256(5):772-8; discussion 778-9. doi: 10.1097/SLA.0b013e3182734423. PMID: 23095621.)
5) In the “chemotherapy sequencing” paragraph (line 227), first you state that preoperative treatment allows to test tumor biology eventually avoiding non-beneficial surgery in case of progression. Shortly after (line 235) you state that in borderline resectable CRLM neoadjuvant chemotherapy may lead to progression into unresectability, thus in this specific setting maybe adjuvant treatment only would be a better option, potentially creating confusion for the reader. This introductive paragraph should be clarified: there is preoperative/neoadjuvant/induction treatment (in case of upfront resectable or borderline resectable disease), and conversion treatment (initially unresectable CRLM). While the latter scenario leaves no room for discussion, in the setting of upfront resectable and borderline resectable disease the use of preoperative chemotherapy is much more debated (and you correctly describe it). Then there is post-operative/adjuvant treatment (see below).
6) In the paragraph “adjuvant therapy” (line 240) you do not mention the JCOG0603 trial from Japan (Kanemitsu Y, Shimizu Y, Mizusawa J, Inaba Y, Hamaguchi T, Shida D, Ohue M, Komori K, Shiomi A, Shiozawa M, Watanabe J, Suto T, Kinugasa Y, Takii Y, Bando H, Kobatake T, Inomata M, Shimada Y, Katayama H, Fukuda H; JCOG Colorectal Cancer Study Group. Hepatectomy Followed by mFOLFOX6 Versus Hepatectomy Alone for Liver-Only Metastatic Colorectal Cancer (JCOG0603): A Phase II or III Randomized Controlled Trial. J Clin Oncol. 2021 Dec 1;39(34):3789-3799. doi: 10.1200/JCO.21.01032. Epub 2021 Sep 14. PMID: 34520230.). This trial (to my knowledge) is the most recent available addressing the use of post-operative systemic treatment in CRLM patients and confirmed an advantage in disease-free survival (but not in overall survival) for hepatectomy followed by mFOLFOX6 compared to hepatectomy alone. This trial was one of the bases for ASCO 2023 guidelines on metastatic colorectal cancer (see Table 11 of ASCO 2023 guidelines) and I believe it should be mentioned. That said, lots of retrospective studies reported better OS with the use of adjuvant treatment compared to hepatectomy alone, maybe you could include in the discussion some recent meta-analysis.
7) “Perioperative therapy” paragraph: according to what above mentioned, the initial statement (line 260) could be too strong (there is level 1b evidence in DFS, but not in OS).
8) Line 335: you state that in EPOC trial (EORTC 40983) by Nordlinger, et al. did not include data regarding chemotherapy-related toxicity. This seems to me incorrect, such data are provided in Table 2 and 3 of the trial. Please correct.
9) Line 419: the correct name is “AZOULAY”, please correct.
10) “Resection margin” paragraph (line 469), I would suggest moving the part related to R1 vascular (Viganò, Torzilli) at the end of this paragraph instead than in the “parenchymal-sparing surgery” paragraph. I agree that R1vasc is one of the elements of PSS but it is worthy of mention when you discuss about the length of the margins, again possibly mentioning the quality of the evidence available on R1vasc. Similarly, I wonder if E-OSH (line 648) would be better discussed following TSH and ALPPS rather than in PSS section, however, this last one is just a suggestion.
11) “Robotic liver resection” paragraph (line 697). I suggest to clarify which is the level of evidence currently available, particularly because – in contrast to laparoscopic approach – there is no randomized trials available.
Thanks for addressing the abovementioned points, I wish to thank once again the authors for their effort in elaborating this comprehensive manuscript.
This narrative review aimed to discuss the aspects of the contemporary peri- and post-operative management of patients with colorectal liver metastases (CRLM). The manuscript is well written and very comprehensive; it addressed all the most important elements which must be considered when dealing with CRLM patients.
Some issues should be kindly addressed:
1) In table 1 among the “anatomic factors of resectability”, the ability to achieve a R0 resection could be added to easier the reader’s comprehension (exceptions to this golden rule are well described in other sections of the manuscript).
2) At line 141, you correctly state that the old risk scores model considered clinicopathologic factors but not biological ones. It would be useful to underline another limitation of such risk scores (like the Fong one), which is that those scores were developed in an era when effective chemotherapy still not existed (FOLFOX introduced in clinical practice in the early 2000’s), and that almost none of the patients included in those series received preoperative/neoadjuvant treatments. Nowadays many CRLM patients receive preoperative systemic treatment (based on oncologic more than anatomic risk factors), questioning the applicability of the old risk scores. Lastly, non-response or progression during preoperative chemotherapy could be considered a surrogate marker of biological aggressiveness of the disease.
3) In the section “operative sequencing for synchronous disease” (line 157), you should precise that irrespective of the decision to which site resect first, most of the patients should receive preoperative treatment (i.e., neoadjuvant or conversion intent). You should also precise in which conditions an upfront resective approach (and not following a systemic treatment) could be considered (symptomatic primary, obstruction, bleeding, etc.).
4) Specifically regarding the “liver-first” approach, would you please cite the first landmark paper describing this strategy (Mentha G, Majno PE, Andres A, Rubbia-Brandt L, Morel P, Roth AD. Neoadjuvant chemotherapy and resection of advanced synchronous liver metastases before treatment of the colorectal primary. Br J Surg. 2006 Jul;93(7):872-8. doi: 10.1002/bjs.5346. PMID: 16671066.), and the subsequent work validating it on a large scale (Andres A, Toso C, Adam R, Barroso E, Hubert C, Capussotti L, Gerstel E, Roth A, Majno PE, Mentha G. A survival analysis of the liver-first reversed management of advanced simultaneous colorectal liver metastases: a LiverMetSurvey-based study. Ann Surg. 2012 Nov;256(5):772-8; discussion 778-9. doi: 10.1097/SLA.0b013e3182734423. PMID: 23095621.)
5) In the “chemotherapy sequencing” paragraph (line 227), first you state that preoperative treatment allows to test tumor biology eventually avoiding non-beneficial surgery in case of progression. Shortly after (line 235) you state that in borderline resectable CRLM neoadjuvant chemotherapy may lead to progression into unresectability, thus in this specific setting maybe adjuvant treatment only would be a better option, potentially creating confusion for the reader. This introductive paragraph should be clarified: there is preoperative/neoadjuvant/induction treatment (in case of upfront resectable or borderline resectable disease), and conversion treatment (initially unresectable CRLM). While the latter scenario leaves no room for discussion, in the setting of upfront resectable and borderline resectable disease the use of preoperative chemotherapy is much more debated (and you correctly describe it). Then there is post-operative/adjuvant treatment (see below).
6) In the paragraph “adjuvant therapy” (line 240) you do not mention the JCOG0603 trial from Japan (Kanemitsu Y, Shimizu Y, Mizusawa J, Inaba Y, Hamaguchi T, Shida D, Ohue M, Komori K, Shiomi A, Shiozawa M, Watanabe J, Suto T, Kinugasa Y, Takii Y, Bando H, Kobatake T, Inomata M, Shimada Y, Katayama H, Fukuda H; JCOG Colorectal Cancer Study Group. Hepatectomy Followed by mFOLFOX6 Versus Hepatectomy Alone for Liver-Only Metastatic Colorectal Cancer (JCOG0603): A Phase II or III Randomized Controlled Trial. J Clin Oncol. 2021 Dec 1;39(34):3789-3799. doi: 10.1200/JCO.21.01032. Epub 2021 Sep 14. PMID: 34520230.). This trial (to my knowledge) is the most recent available addressing the use of post-operative systemic treatment in CRLM patients and confirmed an advantage in disease-free survival (but not in overall survival) for hepatectomy followed by mFOLFOX6 compared to hepatectomy alone. This trial was one of the bases for ASCO 2023 guidelines on metastatic colorectal cancer (see Table 11 of ASCO 2023 guidelines) and I believe it should be mentioned. That said, lots of retrospective studies reported better OS with the use of adjuvant treatment compared to hepatectomy alone, maybe you could include in the discussion some recent meta-analysis.
7) “Perioperative therapy” paragraph: according to what above mentioned, the initial statement (line 260) could be too strong (there is level 1b evidence in DFS, but not in OS).
8) Line 335: you state that in EPOC trial (EORTC 40983) by Nordlinger, et al. did not include data regarding chemotherapy-related toxicity. This seems to me incorrect, such data are provided in Table 2 and 3 of the trial. Please correct.
9) Line 419: the correct name is “AZOULAY”, please correct.
10) “Resection margin” paragraph (line 469), I would suggest moving the part related to R1 vascular (Viganò, Torzilli) at the end of this paragraph instead than in the “parenchymal-sparing surgery” paragraph. I agree that R1vasc is one of the elements of PSS but it is worthy of mention when you discuss about the length of the margins, again possibly mentioning the quality of the evidence available on R1vasc. Similarly, I wonder if E-OSH (line 648) would be better discussed following TSH and ALPPS rather than in PSS section, however, this last one is just a suggestion.
11) “Robotic liver resection” paragraph (line 697). I suggest to clarify which is the level of evidence currently available, particularly because – in contrast to laparoscopic approach – there is no randomized trials available.
Thanks for addressing the abovementioned points, I wish to thank once again the authors for their effort in elaborating this comprehensive manuscript.
This narrative review aimed to discuss the aspects of the contemporary peri- and post-operative management of patients with colorectal liver metastases (CRLM). The manuscript is well written and very comprehensive; it addressed all the most important elements which must be considered when dealing with CRLM patients.
Some issues should be kindly addressed:
1) In table 1 among the “anatomic factors of resectability”, the ability to achieve a R0 resection could be added to easier the reader’s comprehension (exceptions to this golden rule are well described in other sections of the manuscript).
2) At line 141, you correctly state that the old risk scores model considered clinicopathologic factors but not biological ones. It would be useful to underline another limitation of such risk scores (like the Fong one), which is that those scores were developed in an era when effective chemotherapy still not existed (FOLFOX introduced in clinical practice in the early 2000’s), and that almost none of the patients included in those series received preoperative/neoadjuvant treatments. Nowadays many CRLM patients receive preoperative systemic treatment (based on oncologic more than anatomic risk factors), questioning the applicability of the old risk scores. Lastly, non-response or progression during preoperative chemotherapy could be considered a surrogate marker of biological aggressiveness of the disease.
3) In the section “operative sequencing for synchronous disease” (line 157), you should precise that irrespective of the decision to which site resect first, most of the patients should receive preoperative treatment (i.e., neoadjuvant or conversion intent). You should also precise in which conditions an upfront resective approach (and not following a systemic treatment) could be considered (symptomatic primary, obstruction, bleeding, etc.).
4) Specifically regarding the “liver-first” approach, would you please cite the first landmark paper describing this strategy (Mentha G, Majno PE, Andres A, Rubbia-Brandt L, Morel P, Roth AD. Neoadjuvant chemotherapy and resection of advanced synchronous liver metastases before treatment of the colorectal primary. Br J Surg. 2006 Jul;93(7):872-8. doi: 10.1002/bjs.5346. PMID: 16671066.), and the subsequent work validating it on a large scale (Andres A, Toso C, Adam R, Barroso E, Hubert C, Capussotti L, Gerstel E, Roth A, Majno PE, Mentha G. A survival analysis of the liver-first reversed management of advanced simultaneous colorectal liver metastases: a LiverMetSurvey-based study. Ann Surg. 2012 Nov;256(5):772-8; discussion 778-9. doi: 10.1097/SLA.0b013e3182734423. PMID: 23095621.)
5) In the “chemotherapy sequencing” paragraph (line 227), first you state that preoperative treatment allows to test tumor biology eventually avoiding non-beneficial surgery in case of progression. Shortly after (line 235) you state that in borderline resectable CRLM neoadjuvant chemotherapy may lead to progression into unresectability, thus in this specific setting maybe adjuvant treatment only would be a better option, potentially creating confusion for the reader. This introductive paragraph should be clarified: there is preoperative/neoadjuvant/induction treatment (in case of upfront resectable or borderline resectable disease), and conversion treatment (initially unresectable CRLM). While the latter scenario leaves no room for discussion, in the setting of upfront resectable and borderline resectable disease the use of preoperative chemotherapy is much more debated (and you correctly describe it). Then there is post-operative/adjuvant treatment (see below).
6) In the paragraph “adjuvant therapy” (line 240) you do not mention the JCOG0603 trial from Japan (Kanemitsu Y, Shimizu Y, Mizusawa J, Inaba Y, Hamaguchi T, Shida D, Ohue M, Komori K, Shiomi A, Shiozawa M, Watanabe J, Suto T, Kinugasa Y, Takii Y, Bando H, Kobatake T, Inomata M, Shimada Y, Katayama H, Fukuda H; JCOG Colorectal Cancer Study Group. Hepatectomy Followed by mFOLFOX6 Versus Hepatectomy Alone for Liver-Only Metastatic Colorectal Cancer (JCOG0603): A Phase II or III Randomized Controlled Trial. J Clin Oncol. 2021 Dec 1;39(34):3789-3799. doi: 10.1200/JCO.21.01032. Epub 2021 Sep 14. PMID: 34520230.). This trial (to my knowledge) is the most recent available addressing the use of post-operative systemic treatment in CRLM patients and confirmed an advantage in disease-free survival (but not in overall survival) for hepatectomy followed by mFOLFOX6 compared to hepatectomy alone. This trial was one of the bases for ASCO 2023 guidelines on metastatic colorectal cancer (see Table 11 of ASCO 2023 guidelines) and I believe it should be mentioned. That said, lots of retrospective studies reported better OS with the use of adjuvant treatment compared to hepatectomy alone, maybe you could include in the discussion some recent meta-analysis.
7) “Perioperative therapy” paragraph: according to what above mentioned, the initial statement (line 260) could be too strong (there is level 1b evidence in DFS, but not in OS).
8) Line 335: you state that in EPOC trial (EORTC 40983) by Nordlinger, et al. did not include data regarding chemotherapy-related toxicity. This seems to me incorrect, such data are provided in Table 2 and 3 of the trial. Please correct.
9) Line 419: the correct name is “AZOULAY”, please correct.
10) “Resection margin” paragraph (line 469), I would suggest moving the part related to R1 vascular (Viganò, Torzilli) at the end of this paragraph instead than in the “parenchymal-sparing surgery” paragraph. I agree that R1vasc is one of the elements of PSS but it is worthy of mention when you discuss about the length of the margins, again possibly mentioning the quality of the evidence available on R1vasc. Similarly, I wonder if E-OSH (line 648) would be better discussed following TSH and ALPPS rather than in PSS section, however, this last one is just a suggestion.
11) “Robotic liver resection” paragraph (line 697). I suggest to clarify which is the level of evidence currently available, particularly because – in contrast to laparoscopic approach – there is no randomized trials available.
Thanks for addressing the abovementioned points, I wish to thank once again the authors for their effort in elaborating this comprehensive manuscript.
